

**Cluster Dynamics-based Parameterization for Sulfuric Acid-Dimethylamine**
**Nucleation: Comparison and Selection through Box- and Three-Dimensional-**
**Modeling**
Jiewen Shen[1,2], Bin Zhao[1,2], Shuxiao Wang[1,2,*], An Ning[3], Yuyang Li[2], Runlong Cai[4],
Da Gao[1,2], Biwu Chu[5,6], Yang Gao[7], Manish Shrivastava[8], Jingkun Jiang[2], Xiuhui
Zhang[3], Hong He[5,6]
[1]*State Key Joint Laboratory of Environment Simulation and Pollution Control, School*
*of Environment, Tsinghua University, Beijing, 100084, China*
[2]*State Environmental Protection Key Laboratory of Sources and Control of Air*
*Pollution Complex, Beijing, 100084, China*
[3]*Key Laboratory of Cluster Science, Ministry of Education of China, School of*
*Chemistry and Chemical Engineering, Beijing Institute of Technology, Beijing, 100081,*
*China*
[4]*Shanghai Key Laboratory of Atmospheric Particle Pollution and Prevention (LAP[3]),*
*Department of Environmental Science & Engineering, Fudan University, Shanghai,*
*200438, China*
[5]*State Key Joint Laboratory of Environment Simulation and Pollution Control,*
*Research Center for Eco-Environmental Sciences, Chinese Academy of Sciences,*
*Beijing 100085, China*
[6]*College of Resources and Environment, University of Chinese Academy of Sciences,*
*Beijing 100049, China*
[7]*Key Laboratory of Marine Environment and Ecology, Ministry of Education, Ocean*
*University of China, Qingdao 266100, China*
[8]*Pacific Northwest National Laboratory, Richland, Washington, USA*
*Correspondence to: Shuxiao Wang (shxwang@tsinghua.edu.cn)



**ABSTRACT**
Clustering of gaseous sulfuric acid (SA) enhanced by dimethylamine (DMA) is a
major mechanism for new particle formation (NPF) in polluted atmospheres. However,
uncertainty remains regarding the SA-DMA nucleation parameterization that
reasonably represents cluster dynamics and is applicable across various atmospheric
conditions. This uncertainty hinders accurate three-dimensional (3-D) modeling of NPF
and subsequent assessment of its environmental and climatic impacts. Here we
extensively compare different cluster dynamics-based parameterizations for SA-DMA
nucleation and identify the most reliable one through a combination of box-model
simulations, 3-D modeling, and in-situ observations. Results show that the
parameterization derived from Atmospheric Cluster Dynamic Code (ACDC)
simulations, incorporating the latest theoretical insights (DLPNO-CCSD(T)/aug-cc-
pVTZ//ωB97X-D/6-311++G(3df,3pd) level of theory) and adequate representation of
cluster dynamics, exhibits dependable performance in 3-D NPF simulation for both
winter and summer conditions in Beijing and shows promise for application in diverse
atmospheric conditions. Another ACDC-derived parameterization, replacing the level
of theory with RI-CC2/aug-cc-pV(T+d)Z//M06-2X/6–311++G(3df,3pd), also performs
well in NPF modeling at relatively low temperatures around 280 K but exhibits
limitations at higher temperatures due to inappropriate representation of SA-DMA
cluster thermodynamics. Additionally, a previously reported parameterization
incorporating simplifications is applicable for simulating NPF in polluted atmospheres
but tends to overestimate particle formation rates under conditions of elevated
temperature (> ~300 K) and low condensation sink (< ~3×10$^{-3}$ s$^{-1}$). Our findings
highlight the applicability of the new ACDC-derived parameterization, which couples
the latest SA-DMA nucleation theory and holistic cluster dynamics, in 3-D NPF
modeling. The ACDC-derived parameterization framework provides valuable reference
for developing parameterizations for other nucleation systems.




**GRAPHICAL ABSTRACT**

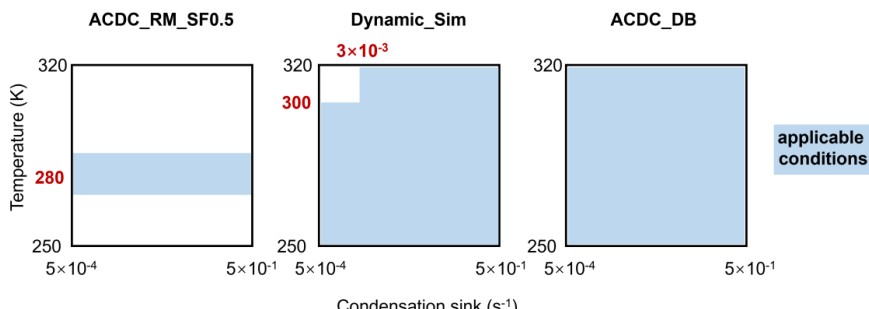

**ACDC_RM_SF0.5:**
    ACDC-derived parameterization with traditional theoretical approach
**Dynamic_Sim:**
    reported parameterization with simplifications in cluster dynamics
**ACDC_DB:**
    ACDC-derived parameterization with latest theoretical approach





## 1 INTRODUCTION

Atmospheric aerosols have significant impacts on visibility, human health, and
global climate (Gordon et al., 2016). New Particle Formation (NPF) is the predominant
source of global aerosol population, with nucleation being the key stage of the gas-to-
particle transformation (Zhao et al., 2020; Almeida et al., 2013). In polluted regions
such as urban China, compelling evidence indicates that sulfuric acid (SA)-driven
nucleation enhanced by dimethylamine (DMA) can generate thermodynamically stable
SA-DMA clusters and lead to high particle formation rates close to kinetic limit of SA
clustering, which is responsible for the observed intensive NPF events (Cai et al., 2021;
Yao et al., 2018). Meanwhile, it has been demonstrated that variations in atmospheric
conditions, including condensation sinks (CS) arising from background aerosols, along
with temperature ($T$), can exert profound impacts on the cluster dynamics of SA-DMA
nucleation by varying the particle formation rates across several orders of magnitude
(Cai et al., 2021; Deng et al., 2020). Given that complex interactions exist among
various gaseous precursors, molecular clusters, and pre-existing aerosols during
nucleation, reasonable representation of the cluster dynamics of SA-DMA nucleation
in three-dimensional (3-D) models is important for 3-D NPF modeling and subsequent
assessment of its impacts on environment and climate.
Empirical models in form of power law functions have been extensively utilized to
examine how particle formation rates respond to precursor concentrations (Semeniuk
and Dastoor, 2018). Through parameter fitting, these empirical models can effectively
reproduce the particle formation rates observed in both laboratory experiments and field
measurements (Kulmala et al., 2006; Riccobono et al., 2014; Semeniuk and Dastoor,
2018). Subsequently, they can be integrated into 3-D models for regional or global NPF
simulations. Bergman et al. (2015) and Dunne et al. (2016) have simulated SA-DMA
nucleation utilizing global models, which incorporate empirical equations derived from
experimental data obtained from CLOUD chamber or flow tube experiments. These
parameterization schemes successfully characterize the response of particle formation
rates to precursor concentrations, however, they fail to account for dependencies on $T$
and CS due to the ignorance of explicit cluster dynamics. As a result, they are identified
to be inadequate for accurately reproducing NPF events in winter Beijing (Li et al.,
91 2023).

We recently developed an analytical equation for SA-DMA nucleation
parameterization based on detailed cluster dynamics simulations (abbreviated as
Dynamic_Sim) (Li et al., 2023c). To derive an explicit equation, several simplifications
have been made in Dynamic_Sim, including 1) only $(SA)_k(DMA)_k$ ($k$ = 1-4) and
$(SA)_2(DMA)_1$ clusters are considered; 2) clusters larger than $(SA)_1(DMA)_1$ are
regarded stable with no evaporation; and 3) $(SA)_4(DMA)_4$ cluster is the only terminal
cluster in calculating particle formation rates. Subsequent applications in 3-D modeling
have demonstrated significantly improved performance of Dynamic_Sim compared to
previous data-fitting parameterizations in simulating the particle formation rates, the
evolution of particle number size distributions (PNSDs), and NPF events in winter



Beijing. However, the efficacy of Dynamic_Sim in NPF simulation has yet to be
assessed under varying atmospheric conditions, such as the summer season
characterized by relatively higher $T$ and lower CS compared to winter. Moreover, the
impacts of simplifications made in the derivation of Dynamic_Sim on 3-D NPF
simulation under different atmospheric conditions remain unclear.
Atmospheric Cluster Dynamics Code (ACDC) is a flexible box model for
simulating cluster dynamics and particle formation rates (Mcgrath et al., 2012). In
addition to representing $T$- and CS- dependencies for particle formation rate as
Dynamic_Sim, ACDC considers the source/sink terms of all given molecules/clusters
within a nucleation system without simplifications of the clustering processes. By
integrating quantum chemical calculations with ACDC, Almeida et al. (2013)
discovered that the simulated SA-DMA nucleation provides valuable insights for
interpreting the measurements from the CLOUD chamber experiments. Similarly, Lu
et al. (2020) demonstrated that ACDC coupled with quantum chemistry calculations
can effectively reproduce the particle formation rates observed in urban Shanghai.
While ACDC has been extensively utilized in box modeling (Almeida et al., 2013; Lu
et al., 2020; Yang et al., 2021), its potential for deriving parameterizations for 3-D
models has not been explored in previous studies. Furthermore, ACDC program in
modeling the nucleation process is highly reliant on specific thermodynamic data for
the molecular clusters of interest, which are primarily obtained through quantum
chemical calculations (Elm et al., 2020). The uncertainty surrounding the influence of
different quantum chemical calculation approaches adds additional complexity to the
application of ACDC-derived parameterization in 3-D NPF modeling.
This study aims to compare different cluster dynamic-based parameterizations for
SA-DMA nucleation and identify the robust one applicable for 3-D models. We
introduced new parameterizations developed using the ACDC program, incorporating
various quantum chemical calculations. Different cluster dynamic-based
parameterizations, including ACDC-derived ones as well as Dynamic_Sim, are
comprehensively compared and evaluated through a combination of box-model
simulations, 3-D modeling, and in-situ observational data. Our findings reveal that by
incorporating the latest theoretical understanding and complete representation of cluster
dynamics, ACDC-derived parameterization demonstrates reliable performance in 3-D
NPF simulation for both winter and summer conditions in Beijing and exhibits potential
applicability in diverse atmospheric conditions. The study sheds light on the impacts of
employing various simplifications in cluster dynamics and different theoretical
approaches in deriving parameterizations on NPF simulation. In addition to
contributing to the precise simulation of SA-DMA nucleation and the quantification of
its environmental and climatic effects, this study provides valuable references for
simulating other nucleation mechanisms in 3-D models.
**2 METHODS**
**2.1 Configurations of ACDC**
Here, $(SA)_m(DMA)_n$ clusters ($0 < n \leqslant m \leqslant 3$, $n$ and $m$ represent the number of SA



and DMA molecules in a cluster) are used to build the ACDC-derived parameterizations
for SA-DMA nucleation due to their reported much higher stability compared to those
containing more DMA molecules than SA molecules (Xie et al., 2017). The
conformations and thermodynamics of SA-DMA clusters are taken from our other
study (Ning et al., 2024). Briefly, the conformations of selected clusters are taken from
the reported global minima from Li et al. (2020), and the key thermodynamic data for
ACDC, Gibbs free energy change ($\Delta G$), are recalculated at the DLPNO-CCSD(T)/aug-
cc-pVTZ//$\omega$B97X-D/6-311++G(3df,3pd) level of theory. Based on benchmark studies
(Elm et al., 2020), this level of theory provides dependable thermodynamic insights into
molecular clusters during nucleation and represents the latest theoretical approach. In
addition, the rotational symmetry is consistently considered in quantum calculations
following Besel et al. (2020). Following most previous ACDC simulation studies (Xie
et al., 2017; Elm et al., 2020; Ning et al., 2020), $(SA)_4(DMA)_3$ and $(SA)_4(DMA)_4$
clusters are defined as the boundary conditions, i.e. the clusters fluxing out the
simulated system and participating in subsequent growth in ACDC simulations,
considering their high stability. Since clusters containing SA tetramers are estimated to
have an electrical mobility diameter of 1.4 nm (Cai et al., 2023; Jen et al., 2014; Thomas
et al., 2016), the formation rates of $(SA)_4(DMA)_3$ and $(SA)_4(DMA)_4$ clusters are
therefore deemed as the particle formation rates at 1.4 nm ($J_{1.4}$). Size-dependent
coagulation sink (CoagS) is counted for each SA-DMA cluster which is consistent with
Dynamic_Sim (Li et al., 2023c):
$$CoagS_i = CS\ (\frac{V_i}{V_1})^{-\frac{1.7}{3}}$$
where $V_i$ and $V_1$ (m$^3$) represent the volume of cluster $i$ and SA molecule, respectively.
The power-law exponent of -1.7 is selected according to typical range in the atmosphere
(Lehtinen et al., 2007). In addition, enhancement for collision processes from Van de
Waals forces is also considered. We refer to the ACDC-derived parameterization in
coupling the DLPNO-CCSD(T)/aug-cc-pVTZ//$\omega$B97X-D/6-311++G(3df,3pd) level of
theory and adequate cluster dynamics as ACDC_DB, which is established as the base-
case for our discussion of other cluster dynamics-based parameterizations.
In addition to the direct comparison of ACDC_DB to Dynamic_Sim, additional test
parameterizations combining ACDC_DB and three simplifications within
Dynamic_Sim are established and compared with ACDC_DB to further probe the
impacts of these simplifications on NPF simulations. The configurations of all
parameterizations are detailed in Table 1. It should be noted that when all
simplifications are applied on ACDC_DB, Dynamic_Sim still predicts higher $J_{1.4}$
compared to ACDC_DB (Figure S1A). This is because the $\Delta G$ value of the initial
$(SA)_1(DMA)_1$ cluster at 298.15 K used in Dynamic_Sim, which is taken from Myllys
et al. (Myllys et al., 2019), is slightly lower than that used in ACDC_DB (-13.5 kcal
mol$^{-1}$ for Dynamic_Sim and -12.9 kcal mol$^{-1}$ for ACDC_DB) (Ning et al., 2024), even
though both parameterizations employ the quantum chemical calculation method of
DLPNO-CCSD(T). Possible reasons for the discrepancy include the utilization of a



larger basis set (3-zeta 6-311++G(3df,3pd)) and higher convergence criteria (Tight
PNO + Tight SCF) in this study compared to that in Myllys et al.. Aligning the $\Delta G$ for
$(SA)_1(DMA)_1$ cluster in Dynamic_Sim with that of ACDC leads to a high consistency
in the predicted $J_{1.4}$ between the two approaches (Figure S1B). The uncertainty of $\Delta G$
used in Dynamic_Sim is discussed in our previous study (Li et al., 2023c) and here we
mainly focus on the impacts of simplifications in Dynamic_Sim.
While the DLPNO-CCSD(T)/aug-cc-pVTZ//ωB97X-D/6-311++G(3df,3pd) level
of theory yields reasonable cluster thermodynamics, quantum chemistry calculations
employing the RI-CC2 method predicting lower $\Delta G$ for cluster formation (stronger
binding between molecules within clusters), has been widely used in conjunction with
ACDC to interpret experimental and observed particle formation rates in previous
studies (Almeida et al., 2013; Kürten et al., 2018; Ning et al., 2020). The prevalent
combination used with the RI-CC2 method is RI-CC2/aug-cc-pV(T+d)Z//M06-2X/6-
311++G(3df,3pd) level of theory (Lu et al., 2020; Liu et al., 2021; Ning et al., 2022;
Ning and Zhang, 2022; Liu et al., 2019). Based on Elm's work, compared to DLPNO-
CCSD(T)/aug-cc-pVTZ//ωB97X-D/6-311++G(3df,3pd), the differences in predicted
cluster binding energies primarily stem from discrepancies between DLPNO-CCSD(T)
and RI-CC2 in single-point energy calculations, while the ωB97X-D and M06-2X
functionals exhibit similar performance (Elm et al., 2013; Elm et al., 2020). Also, in
previous studies the RI-CC2 method combined with ACDC was consistently
accompanied by application of a sticking factor (SF) of 0.5 in treating collision
processes (Almeida et al., 2013; Lu et al., 2020). However, it is noteworthy that,
according to Stolzenburg et al.'s work (Stolzenburg et al., 2020), the SF of the neutral
SA-DMA cluster system should be unity. Here, we refer to the traditional theoretical
approach as employing the RI-CC2/aug-cc-pV(T+d)Z//M06-2X/6-311++G(3df,3pd)
level of theory and incorporating the SF of 0.5 in collision processes. An ACDC-derived
parameterization coupling the traditional theoretical approach is established to assess
the effectiveness of the traditional method in NPF simulation (ACDC_RM_SF0.5).
Except for the varied thermodynamic inputs and SF, the remaining configurations of
ACDC_RM_SF0.5 are identical to ACDC_DB. Additionally, we establish a test
parameterization coupling RI-CC2/aug-cc-pV(T+d)Z//M06-2X/6-311++G(3df,3pd)
level of theory with an SF of unity (ACDC_RM) to evaluate the impact solely arising
from the quantum chemical calculation method. Note that SF of unity is applied to all
parameterizations in this study except for the ACDC_RM_SF0.5.
To quantify the differences in simulating $J_{1.4}$ among different cluster dynamics-
based parameterizations compared to our base-case ACDC_DB, we introduce a
parameter $R$:
$$R_X = \frac{\sum_i^n (X_i / \text{ACDC\_DB}_i)}{n}$$

where $\text{ACDC\_DB}_i$ and $X_i$ denote the simulated $J_{1.4}$ by the base-case ACDC_DB and
another specific parameterization X, respectively, given the input scenarios of $i$ (a set
of input values for $T$, CS, concentration of SA ([SA]) and DMA ([DMA])), and $n$





signifies the total number of input scenarios.

**Table 1.** Summary of various cluster dynamics-based parameterizations of SA-DMA
nucleation in this study (main parameterizations are in bold, while test ones in regular)

| Case | Description |
|---|---|
| **Dynamic_Sim** | Reported parameterization from Li et al. 2023 combining the simplifications in boundary conditions, cluster evaporations, and cluster number |
| **ACDC_DB** | ACDC-derived parameterization coupling DLPNO-CCSD(T)/aug-cc-pVTZ//ωB97X-D/6-311++G(3df,3pd) level of theory, namely the latest theoretical approach |
| ACDC_DB_BC | ACDC-derived parameterization coupling DLPNO-CCSD(T)/aug-cc-pVTZ//ωB97X-D/6-311++G(3df,3pd) level of theory and simplification in boundary conditions (only $(SA)_4(DMA)_4$ cluster is set as boundary condition) |
| ACDC_DB_CE | ACDC-derived parameterization coupling DLPNO-CCSD(T)/aug-cc-pVTZ//ωB97X-D/6-311++G(3df,3pd) level of theory and simplification in cluster evaporations (the evaporation rates of $(SA)_k(DMA)_k$ ($k$ = 2-3) and $(SA)_2(DMA)_1$ clusters are kept zero) |
| ACDC_DB_CN | ACDC-derived parameterization coupling DLPNO-CCSD(T)/aug-cc-pVTZ//ωB97X-D/6-311++G(3df,3pd) level of theory and simplification in cluster number (only $(SA)_k(DMA)_k$ ($k$ = 1-3) and $(SA)_2(DMA)_1$ clusters are involved) |
| **ACDC_RM_SF0.5** | ACDC-derived parameterization coupling RI-CC2/aug-cc-pV(T+d)Z//M06-2X/6-311++G(3df,3pd) level of theory and a SF of 0.5 is applied in collision process, namely the traditional theoretical approach |
| ACDC_RM | ACDC-derived parameterization coupling RI-CC2/aug-cc-pV(T+d)Z//M06-2X/6-311++G(3df,3pd) level of theory and a SF of 1 is applied |


**2.2 Incorporating the ACDC-derived Parameterizations into WRF-Chem/R2D-**
**VBS Model**

Various parameterizations are subsequently implemented in the Weather Research

and Forecasting-Chemistry model (WRF-Chem) integrating an experimentally
constrained Radical Two-Dimensional Volatility Basis Set (2D-VBS) (denoted as
WRF-Chem/R2D-VBS) (Zhao et al., 2020). Incorporating the box-model ACDC into a
3-D model using the explicit mathematical formula, as Dynamic_Sim, proves to be
challenging. Here, we created a four-dimensional look-up table that delineates the



response of $J_{1.4}$ to four input variables ($T$, CS, [SA], and [DMA]) for each ACDC-
derived parameterization (Yu, 2010). The table is derived based on multiple ACDC runs
by varying input variables. The ranges for the input variables correspond to typical
conditions of the atmosphere. Except for $T$, the ranges of variation for all other variables
exceed at least one order of magnitude. Therefore, temperature is assumed to follow
arithmetic uniform distribution, while the other variables are assumed to follow
geometric uniform distribution. Details for the input variables are given in Table S1. In
WRF-Chem/R2D-VBS simulations, $J_{1.4}$ are online calculated by interpolating values
from a look-up table based on real-time input parameters. In our previous study, we
have developed an emission inventory for China and its surrounding regions (Li et al.,
2023c). Here [DMA] is calculated in WRF-Chem/R2D-VBS based on a comprehensive
source-sink representation of DMA. More details of including DMA in WRF-
Chem/R2D-VBS can be found in our previous study (Li et al., 2023c). In addition, a
time-integrated-average [DMA] as well as [SA] of each time step were used to drive
SA-DMA nucleation, since SA-DMA nucleation is accompanied with condensation of
gaseous SA and DMA on pre-existing aerosols simultaneously in the atmosphere.
Besides SA-DMA nucleation, seven other nucleation mechanisms have already
been incorporated in WRF-Chem/R2D-VBS (Zhao et al., 2020), including neutral/ion-
induced $H_2SO_4$-$H_2O$ nucleation, neutral/ion-induced $H_2SO_4$-$NH_3$-$H_2O$ nucleation,
neutral/ion-induced pure organics nucleation, and $H_2SO_4$-organics nucleation. The
organics involved in nucleation are ultralow- and extremely low-volatility organic
compounds (ULVOC and ELVOC) with O:C >0.4. The formation chemistry of ULVOC
and ELVOC from monoterpenes, including autoxidation and dimerization, is traced by
the R2D-VBS framework (Zhao et al., 2020). Note that the impact of the other seven
mechanisms on particle formation rates and particle number concentration is low
compared to SA-DMA as revealed by our previous study (Li et al., 2023c). In WRF-
Chem/R2D-VBS, the evolution of PNSDs from 1nm to 10 μm is treated by MOSAIC
(Model for Simulating Aerosol Interactions and Chemistry) module. The newly formed
1.4 nm particles from SA-DMA nucleation are injected into the smallest size bin (1 -
1.5 nm) of the MOSAIC.
**2.3 Configurations of WRF-Chem/R2D-VBS Model**
The WRF-Chem/R2D-VBS model, incorporating various cluster dynamics-based
SA-DMA nucleation parameterizations, was employed in a simulation over a domain
with a spatial resolution of 27 km. This domain covers eastern Asia, with Beijing
situated close to the center of the simulation area. Details of model configurations can
be found in our previous study (Li et al., 2023c). Briefly, we use the ABaCAS-EI 2017
and IIASA 2015 emission inventories for mainland China and other areas in the domain,
respectively, to represent the anthropogenic emissions (Zheng et al., 2019; Li et al.,
2017; Li et al., 2023b); we use Model of Emissions of Gases and Aerosols from Nature
(MEGAN) v2.04 to calculate the biogenic emissions (Guenther et al., 2006). The
simulation results from the National Center for Atmospheric Research's Community
Atmosphere Model with Chemistry (https://www.acom.ucar.edu/cam-chem/cam-



chem.shtml) is used for the chemical initial and boundary conditions in WRF-
Chem/R2D-VBS simulations.
The simulation period consists of two parts: the winter period, which spans from
January 14 to January 31, 2019, and the summer period, which is from August 18 to
August 31, 2019. Previous observational studies have shown that the particle formation
rates reach their highest and lowest levels during winter and summer in China,
respectively (Deng et al., 2020; Chu et al., 2019). Therefore, periods from these two
seasons are selected as representative simulation periods in this study and the specific
time periods corresponded to those with relatively complete and continuous PNSDs and
$J_{1.4}$ observations. Since observational data for DMA concentration is only available for
the period from January 1, 2019 to January 23, 2019, similar to our other study (Ning
et al., 2024), we performed additional simulation for this period to compare
observational and simulated DMA concentrations. For each season, all the SA-DMA
parameterizations listed in Table 1 were employed for simulation. Among them,
ACDC_DB, Dynamic_Sim, and ACDC_RM_SF0.5 serve as three main
parameterizations, while ACDC_DB_CE, ACDC_DB_BC, ACDC_DB_CN, and
ACDC_RM are set as test cases to investigate the impact of individual simplification
or theoretical approach on NPF simulations. In all comparisons, ACDC_DB is set as a
reference.

**2.4 Ambient Measurements**

In the 3-D simulations, we utilize measured concentrations of nucleation precursors
and PNSDs as a criterion to discuss the model performance with various
parameterizations. The duration of the observational data matches that of the
simulations mentioned above. Detailed descriptions of the observation site and
instruments can be found in our previous research (Deng et al., 2020). Briefly, the
observation site is located on the West Campus of the Beijing University of Chemical
Technology. CI-TOF-MS (chemical ionization time-of-flight mass spectrometer;
Aerodyne Research Inc.) were used to measure the concentrations of SA. Amine
concentrations were measured with a modified TOF-MS using $H_3O^+$ or its clusters as
the reagent ions. PNSDs from 1 nm to 10 μm were measured using a PSD (particle size
distribution) system and a DEG-SMPS (diethyl glycol scanning mobility particle
spectrometer). $J_{1.4}$ derived from observation is calculated employing an improved
aerosol population balance formula (Cai and Jiang, 2017).



## 3 RESULTS AND DISCUSSIONS

### 3.1 Comparison of Different Parameterizations Based on Box-Model Simulations

#### 3.1.1 Comparison between ACDC_DB and Dynamic_Sim

Figure 1 illustrates the comparison between the reported cluster dynamics-based parameterization with simplifications, Dynamic_Sim, and the base-case parameterization ACDC_DB. The comparison is based on a comprehensive dataset that includes over 40,000 box-model simulations for each parameterization, by varying parameters such as [SA] ($1 \times 10^5 - 1 \times 10^8$ molec. cm$^{-3}$), [DMA] ($5 \times 10^6 - 1 \times 10^8$ molec. cm$^{-3}$), CS ($5 \times 10^{-4} - 5 \times 10^{-1}$ s$^{-1}$), and $T$ (250 – 320 K). In most scenarios, $J_{1.4}$ predicted by ACDC_DB and Dynamic_Sim demonstrates deviations within one order of magnitude, with the majority falling within a factor of 3. However, Dynamic_Sim predicts notably higher $J_{1.4}$ than ACDC_DB in scenarios where $T$ exceeds ~300 K and CS is below ~$3\times10^{-3}$ s$^{-1}$, characteristic of a clean atmosphere during summer. The discrepancy in these scenarios elevates the overall $R_{\text{Dynamic\_Sim}}$ up to 17.0. Furthermore, no clear correlation is observed between the differences of the two parameterizations and other input parameters such as [DMA] and [SA] (Figure S2). The differences between parameterizations are attributed to the combined effects of the three simplifications and the lower $\Delta G$ of $(SA)_1(DMA)_1$ cluster in Dynamic_Sim. However, the latter should not be the primary cause for the significant differences of $J_{1.4}$ prediction under high $T$ and low CS conditions, as it typically results in an overestimation within an order of magnitude ($R$=3.3) (Figure S1).

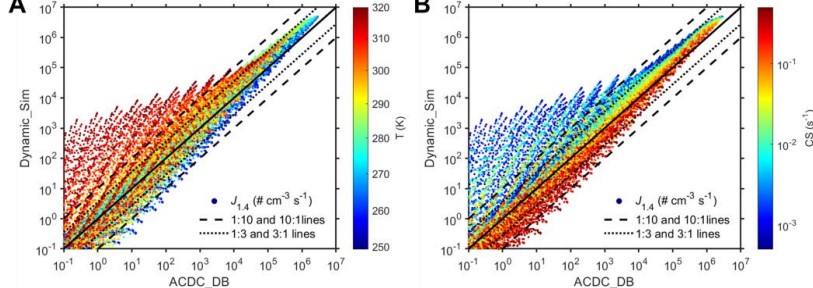

**Figure 1.** Comparison of $J_{1.4}$ predictions between ACDC_DB and Dynamic_Sim correlated with $T$ variation (A) and CS variation (B). Solid dots represent simulated $J_{1.4}$ values, solid lines indicate a 1:1 line, dotted lines correspond to 1:3 and 3:1 lines, and dashed lines represent 1:10 and 10:1 lines.

The impacts of the three simplifications made in Dynamic_Sim are shown in Figure 2. Specifically, the simplification in cluster evaporations tends to elevate the predicted $J_{1.4}$, whereas the simplifications in boundary conditions and cluster number tend to lower them. When applying the simplification in cluster evaporations (clusters larger than $(SA)_1(DMA)_1$ are regarded stable with no evaporation) to ACDC_DB, the predicted $J_{1.4}$ by ACDC_DB_CE only slightly exceed than that of ACDC_DB within a



factor of 3 under conditions where $T < {\sim}290$ K and CS $> {\sim}0.1$ s$^{-1}$. However, the
overestimation of $J_{1.4}$ prediction by ACDC_DB_CE becomes much greater with
increasing $T$ and decreasing CS. The discrepancy between ACDC_DB_CE and
ACDC_DB should be primarily attributed to the pivotal role of $T$ in influencing cluster
evaporation rates (Ortega et al., 2012; Deng et al., 2020). At low $T$, the evaporation
rates of clusters are low enough to allow efficient nucleation, thus whether setting the
concerned SA-DMA clusters to evaporate based on the expected evaporation rates does
not lead to a significant impact on $J_{1.4}$ prediction. However, at high $T$, the evaporation
rates of clusters significantly increase, therefore the simplification in cluster
evaporations within ACDC_DB_CE is likely to predict higher $J_{1.4}$ than those with no
simplification. The impact of simplification in cluster evaporations across varying $T$ is
also found in a nonbranched SA-DMA nucleation scheme from 280 K to 298 K reported
by Li et al. (2023a). Note also that the overestimation of ACDC_DB_CE diminishes as
CS increases (Figure 2D), with CS becoming the primary sink in the nucleation system
and the impact of cluster evaporations becoming less pronounced. This underscores the
connection between the specific deviation arising from simplification in cluster
evaporations and the respective contributions of CS and cluster evaporations to the
overall sink for clusters in nucleation. In addition, the relative independence of the
differences between ACDC_DB_CE and ACDC_DB from variations in precursor
concentrations ([SA] and [DMA]) is similar to that between Dynamic_Sim and
ACDC_DB (Figure S3). Overall, the scenarios where ACDC_DB_CE predicts higher
$J_{1.4}$ than ACDC_DB only occurs under conditions of both high $T$ and low CS (Figure
2A and Figure 2D). The averaged discrepancy between ACDC_DB_CE and
ACDC_DB $R_{\text{ACDC\_DB\_CE}}$ is 22.3, closely resembling $R_{\text{Dynamic\_Sim}}$, indicating that the
simplification in cluster evaporations is a major factor contributing to the difference
between Dynamic_Sim and ACDC_DB.

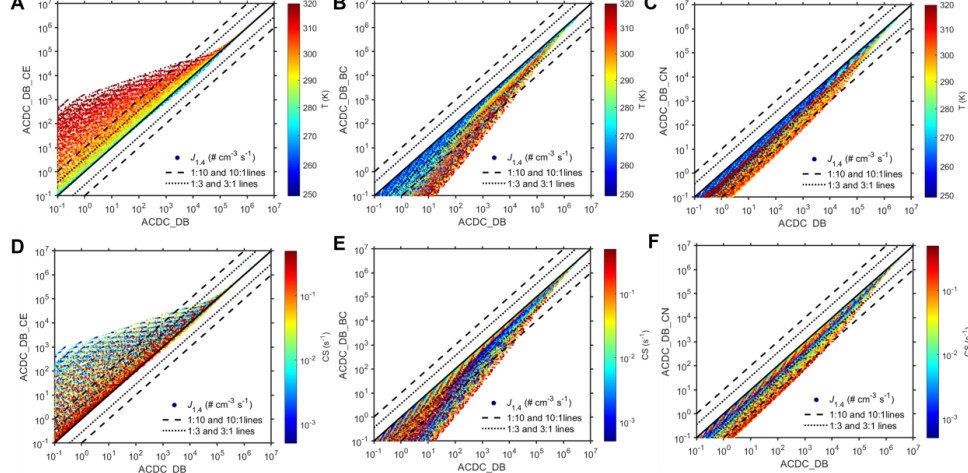

**Figure 2.** Comparison of $J_{1.4}$ predictions between ACDC_DB and test cases including





ACDC_DB_CE (A and D), ACDC_DB_BC (B and E), and ACDC_DB_CN (C and F).
The first row in the panel (A, B and C) is correlated with $T$ variation and the second
row (D, E and F) is correlated with CS variation. Solid dots represent simulated $J_{1.4}$
values, solid lines indicate a 1:1 line, dotted lines correspond to 1:3 and 3:1 lines, and
dashed lines represent 1:10 and 10:1 lines.

The underestimations of ACDC_DB_BC and ACDC_DB_CN in $J_{1.4}$ prediction
compared to base-case ACDC_DB are related to the growth pathways of SA-DMA
clusters. In the original scheme of ACDC_DB, precursor molecules have the flexibility
to pass through any $(SA)_m(DMA)_n$ clusters ($0 < n \leqslant m \leqslant 3$), and terminal 1.4-nm
particles are formed when the clusters grow to $(SA)_4(DMA)_4$ or $(SA)_4(DMA)_3$. As
expected, ACDC_DB_BC, which assumes $(SA)_4(DMA)_4$ cluster as the only boundary
condition with an omission of $(SA)_4(DMA)_3$ cluster, predicts lower $J_{1.4}$ than ACDC_DB.
$(SA)_4(DMA)_3$ and $(SA)_4(DMA)_4$ clusters are primarily formed from $(SA)_3(DMA)_3$
cluster by colliding with a SA molecule and a $(SA)_1(DMA)_1$ cluster, respectively. As
the concentration of $(SA)_1(DMA)_1$ cluster is more sensitive to $T$, we further found that
the discrepancy between ACDC_DB_BC and ACDC_DB becomes more pronounced
with increasing $T$ (Figure 2B). Furthermore, we found no apparent correlation between
the variation of CS and the disparity between ACDC_DB_BC and ACDC_DB (Figure
2E).
In addition to ACDC_DB_BC, ACDC_DB_CN also underestimates $J_{1.4}$ compared
to ACDC_DB with a comparable value (~0.5) of $R_{ACDC\_DB\_CN}$ and $R_{ACDC\_DB\_BC}$. Under
the simplification in cluster number, the formation of 1.4-nm clusters can only occur
through specific pathways, including $(SA)_1(DMA)_1 \rightarrow (SA)_2(DMA)_2 \rightarrow (SA)_3(DMA)_3$
$\rightarrow (SA)_4(DMA)_4/(SA)_4(DMA)_3$, $(SA)_1(DMA)_1 \rightarrow (SA)_2(DMA)_1 \rightarrow (SA)_2(DMA)_2 \rightarrow$
$(SA)_3(DMA)_3 \rightarrow (SA)_4(DMA)_4/(SA)_4(DMA)_3$, or a combination thereof, while other
pathways are restricted. Due to the variability in growth pathways and their
contributions to $J_{1.4}$ under different atmospheric conditions, the difference between
ACDC_DB_CN and ACDC_DB is not strongly correlated with the variations of $T$ and
CS (Figure 2C and Figure 2F). Despite that, while the differences between the two
tested parameterizations (ACDC_DB_BC and ACDC_DB_CN) involving cluster
growth pathways and the original ACDC_DB are not highly correlated with [DMA],
there is a more pronounced correlation with [SA], which implies a more important role
of SA in cluster growth (Figure S4 and Figure S5).
In our previous study, we demonstrated improvements in computing CS- dependent
$J_{1.4}$ of SA-DMA nucleation with the Dynamic_Sim compared to the previous power-
law parameterizations under polluted atmospheric conditions (Li et al., 2023c). Here,
we further show that, based on Dynamic_Sim, the new ACDC_DB with complete
cluster dynamics can more reasonably simulate $J_{1.4}$ under previously less studied
conditions of high $T$ ($> \sim 300$ K) and low CS ($< \sim 3 \times 10^{-3}$ s$^{-1}$), where Dynamic_Sim tends
to produce significant overestimation of $J_{1.4}$. This overestimation is primarily driven by
the simplification in cluster evaporations within Dynamic_Sim. Even though a





comparable performance in $J_{1.4}$ prediction between ACDC_DB and Dynamic_Sim
could be achieved under other ambient conditions, cautions should be made that the
mutual offsetting effect between overestimation and underestimation resulting from
different simplifications in Dynamic_Sim when computing $J_{1.4}$.

**3.1.2 Comparison between ACDC_DB and ACDC_RM_SF0.5**

In Figure 3, ACDC_DB is compared with another main ACDC-derived
parameterization, ACDC_RM_SF0.5, which uses the RI-CC2/aug-cc-pV(T+d)Z//M06-
2X/6-311++G(3df,3pd) level of theory and employs a SF of 0.5 in processing collision.
It can be observed that at lower temperatures (~280 K), ACDC_RM_SF0.5 and
ACDC_DB exhibit similar performance in predicting $J_{1.4}$. However, with higher $T$
(accompanied by lower CS with a slight dependency), $J_{1.4}$ predicted by
ACDC_RM_SF0.5 become higher than that predicted by ACDC_DB, reaching even
several orders of magnitude at the upper limit of the $T$ range (320 K). Furthermore, we
also observed that in scenarios close to the lower limit of the $T$ range (250 K), the $J_{1.4}$
predicted by ACDC_RM_SF0.5 shift from being higher to lower compared to
ACDC_DB.

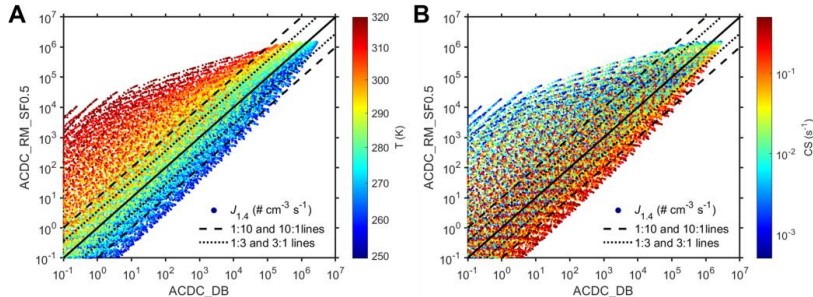

**Figure 3.** Comparison of $J_{1.4}$ predictions between ACDC_DB and ACDC_RM_SF0.5
correlated with $T$ variation (A) and CS variation (B). Solid dots represent simulated $J_{1.4}$
values, solid lines indicate a 1:1 line, dotted lines correspond to 1:3 and 3:1 lines, and
dashed lines represent 1:10 and 10:1 lines.


The distinction between ACDC_RM_SF0.5 and ACDC_DB arises from the
combined effects of variation in quantum chemical calculation method and the
application of the 0.5 SF in collision processing. As depicted in Figure 4, when the SF
in ACDC_RM_SF0.5 is set to unity as in ACDC_DB, the resulting ACDC_RM
parameterization predicts consistently higher $J_{1.4}$ than ACDC_DB. This implies that the
modified quantum chemical calculation method, which results in lower evaporation
rates for clusters within the system compared to ACDC_DB under the same condition,
leads to higher $J_{1.4}$ predictions. The impact from varying quantum chemical calculation
method is akin to that from simplification in cluster evaporations discussed earlier. The
distinction between ACDC_RM and ACDC_DB_CE lies in the fact that the modified
quantum chemical calculation method affects all clusters within the system, whereas





the simplification in cluster evaporations is specific to limited clusters. This contributes
to a much higher $R_{ACDC\_RM}$ (614.5) compared to $R_{ACDC\_DB\_CE}$ (22.3). Despite that,
compared to ACDC_DB, the differences for both ACDC_DB_CE, ACDC_RM, as well
as ACDC_RM_SF0.5 demonstrate similar sensitivity to $T$ (Figure 3A and Figure 4A)
and CS (Figure 3B and Figure 4B) but independence on [SA] (Figure S6A and Figure
S7A) and [DMA] (Figure S6B and Figure S7B). Comparing ACDC_RM_SF0.5 and
ACDC_RM, it can be inferred that the application of a 0.5 SF in collision processes
would result in an underestimation in $J_{1.4}$ prediction. It can be noted that in most
previous studies (Almeida et al., 2013; Kürten et al., 2018; Elm et al., 2020),
comparisons of ACDC simulations using the traditional method and measured particle
formation rates are conducted at around 280 K. At this temperature, all three main
parameterizations of ACDC_RM, ACDC_DB, and Dynamic_Sim tends to yield similar
$J_{1.4}$ predictions and should have consistent applicability in NPF simulation.

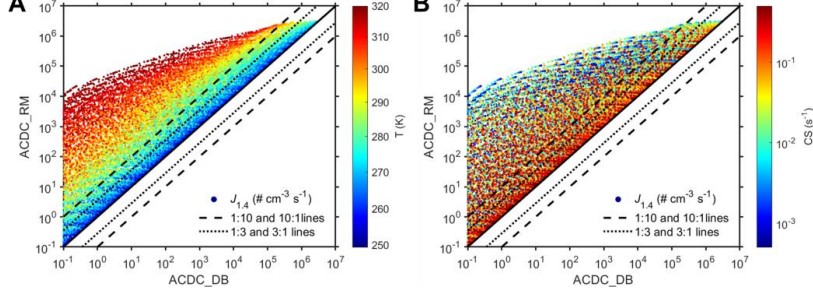

**Figure 4.** Comparison of $J_{1.4}$ predictions between ACDC_DB and ACDC_RM
correlated with $T$ variation (A) and CS variation (B). Solid dots represent simulated $J_{1.4}$
values, solid lines indicate a 1:1 line, dotted lines correspond to 1:3 and 3:1 lines, and
dashed lines represent 1:10 and 10:1 lines.

In summary, based on our base-case parameterization ACDC_DB, the extensive
box model simulations above demonstrate the characteristics and applicability
conditions of different parameterizations. Specifically, Dynamic_Sim is applicable
under most atmospheric conditions with $T < \sim 300$ K and CS $> \sim 3.0 \times 10^{-3}$ s$^{-1}$, while
ACDC_RM_SF0.5 is suitable under conditions with $T$ around 280 K. We further use
reported measurements from well-controlled CLOUD chamber experiments to examine
the characteristics and applicability of these parameterizations (Xiao et al., 2021). As
shown in Figure S8, simulations using three main parameterizations, ACDC_DB,
ACDC_RM, and Dynamic_Sim, correspond well to experimental results at low
temperature ($T = 278$ K), proving the applicability of all three parameterizations at this
temperature. In the experiments with elevated temperature ($T = 293$ K), ACDC_DB and
Dynamic_Sim continues to exhibit similar performance, with simulated results still
corresponding to experimental results. In contrast, ACDC_RM_SF0.5 only shows a
slight $T$-dependence, which is deviated from the measurements. The comparison



between controlled experiments and box-model simulations hence confirms our
conclusions above, and provides a solid basis for further discussions on 3-D simulations
using these parameterizations with constraint from field observations.

**3.2 Comparison of Different Parameterizations Based on 3-D Model Simulations**

Various cluster dynamics-based parameterizations for SA-DMA nucleation were
subsequently integrated into the WRF-Chem/R2D-VBS model. 3-D simulations using
these parameterizations have been conducted for both wintertime and summertime
conditions in Beijing. Given that the concentrations of precursors are crucial input
variables for each parameterization, the simulated and observed concentrations of
[DMA] and [SA] are compared. Figure S9, Figure S10 and Table S2 illustrates good
consistencies in temporal variations and the mean values between simulations and
observations in Beijing. This validates the reliability of our representation of sources
and sinks for nucleating precursors and serves as a foundation for our discussions on
the performances of various parameterizations. In the following sections, we discuss
the results of 3-D NPF simulations in Beijing during winter and summer by employing
different parameterizations. The evaluation of various parameterizations focuses on
their ability to reproduce in situ NPF measurements across different seasons.

**3.2.1 Wintertime Simulations**

Figure 5A and Figure S11A primarily compare the simulated $J_{1.4}$ values from
different parameterizations with those derived from wintertime observations in Beijing,
as $J_{1.4}$ being a key parameter describing NPF events. The performance of Dynamic_Sim
in simulating $J_{1.4}$ during wintertime Beijing has been discussed in our previous study
(Li et al., 2023c). The averaged $J_{1.4}$ simulated by three main parameterizations
(Dynamic_Sim: 64.0 cm$^{-3}$ s$^{-1}$; ACDC_DB: 51.6 cm$^{-3}$ s$^{-1}$; ACDC_RM_SF0.5: 54.5 cm$^{-3}$ s$^{-1}$)
approximate the observation (46.7 cm$^{-3}$ s$^{-1}$). For test cases, however, only
ACDC_DB_CE (55.7 cm$^{-3}$ s$^{-1}$) demonstrates a reasonable representation of $J_{1.4}$. $J_{1.4}$
simulated from ACDC_DB_BC (20.5 cm$^{-3}$ s$^{-1}$) and ACDC_DB_CN (20.8 cm$^{-3}$ s$^{-1}$) are
approximately two times lower than the observed values, while ACDC_RM (226.2 cm$^{-3}$ s$^{-1}$)
is approximately five times higher than the observations.
The performances of different parameterizations on depicting $J_{1.4}$ subsequently
influences their representations of PNSDs evolution and NPF events, which are shown
in Figure 5B. Generally, most parameterizations efficiently reproduce the observed time
evolution of PNSDs and captures NPF events, such as those on 01/20, 01/21, 01/30,
and 01/31, which are characterized by the burst of aerosol number concentrations in
nanometer-sized range. Simulations using ACDC_DB_BC and ACDC_DB_CN result
in lower particle concentrations in the low size range (1-10 nm) during the NPF period
compared to three main parameterizations and the observations, while simulations with
ACDC_RM show higher concentrations, This is consistent with the comparison of $J_{1.4}$
among different parameterizations and further evident by the comparison of averaged
PNSDs in Figure 5C. Notably, when compared to observations, all parameterizations
consistently underestimate the averaged PNSDs within the 2-10 nm range but
overestimate them in the 10-50 nm range. This discrepancy may stem from simplified



assumptions in particle growth simulation, as discussed in our previous study (Li et al., 2023c).

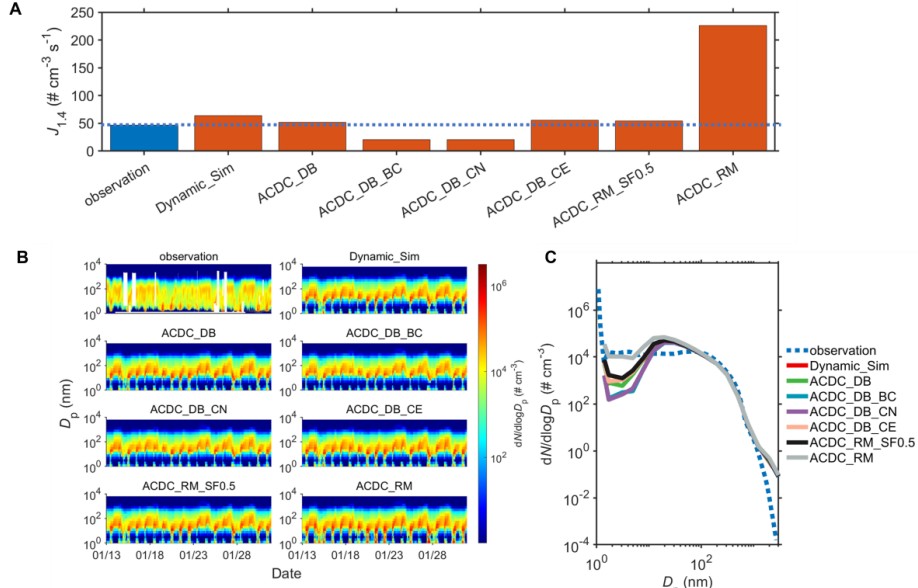

**Figure 5.** Comparison of simulated particle formation rates and particle number size distributions (PNSDs) with observations during January 13, 2019, to January 31, 2019, in Beijing. A represents the averaged particle formation rates during the period, the blue bars and orange bars represent observations and simulations, respectively, while the blue dashed line represents the observed values. Daily maximum values of $J_{1.4}$ are used following Deng et al. (2020); B for the time series of PNSDs; and C for the averaged PNSDs.

The results show the applicability of all three main parameterizations in NPF modeling during wintertime periods. Importantly, the reliability of the new ACDC-derived parameterization based on the latest theoretical approach (ACDC_DB) without simplifications in 3-D NPF simulation, is affirmed. The differences among various parameterizations can be explained by the comprehensive box-model simulations above at corresponding conditions. Compared to ACDC_DB, the $J_{1.4}$ and PNSDs simulated by other two main parameterizations (Dynamic_Sim and ACDC_RM_SF0.5) agree similarly with observations, but for different reasons. In the case of Dynamic_Sim, the simplification in cluster evaporations has minimal impact on NPF simulation since CS is the dominant sink for clusters under the wintertime conditions (averaged $T$ and CS is 274.7 K and $3.3 \times 10^{-2}$ s$^{-1}$, respectively). However, the simplifications in boundary conditions and cluster number lead to the underestimation of the $J_{1.4}$, consequently





lowering the simulated particle number concentrations in 1-100 nm size range due to
the ignorance of clusters contributing to growth. As a result, the agreement of
Dynamic_sim to observations should result from a combination of underestimation due
to simplifications in boundary conditions and cluster number, along with the
compensatory effect of the overestimation caused by lower $\Delta G$ for $(SA)_1(DMA)_1$
cluster. For another main parameterization ACDC_RM_SF0.5, since the test
parameterization ACDC_RM considerably overestimates $J_{1.4}$ and PNSDs compared to
the observations, the general agreement between ACDC_RM_SF0.5 and observations
should be attributed to a balance between reduced kinetic limit through the application
of SF and the compensatory effect of the overestimation caused by inappropriate
representation of cluster thermodynamics.
**3.2.2 Summertime Simulations**
Figure 6 provides additional insight into the performance of various
parameterizations in NPF simulation during summer. It can be noted that there exists a
significant difference in particle formation rates between winter and summer in Beijing.
As shown in Figure 6 and Figure S11B, ACDC_DB and Dynamic_Sim continues to
demonstrate consistent and effective performance in simulating $J_{1.4}$ (within a factor of
2), PNSDs evolution as well as NPF events. However, distinct differences emerge in
the NPF simulation for other parameterizations, including another main
parameterization ACDC_RM_SF0.5. Specifically, in contrast to the good performance
of ACDC_DB and Dynamic_Sim, ACDC_RM_SF0.5, along with the test case
ACDC_RM, exhibits a significant overestimation of $J_{1.4}$, exceeding the observations by
more than 15 times and over two orders of magnitude, respectively. This aligns with
their overestimation of NPF occurrences and particle number concentration in the size
range of 1-100 nm in comparison to observation, with a more pronounced
overestimation for ACDC_RM. Conversely, the test cases of ACDC_DB_BC and
ACDC_DB_CN show an underestimation of averaged $J_{1.4}$ by approximately 4-5 times.
They almost fail to depict NPF events, resulting in a significant underestimation of
number concentrations in the 1-100 nm size range. Simulations using ACDC_DB_CE
notably overestimates $J_{1.4}$ especially on 08/28 – 08/31 (Figure S11B), which results in
an overestimation of averaged $J_{1.4}$ by approximately 4 times compared to the
observations. However, apart from a moderate overestimation in the initial particle size,
we can observe a closer alignment of particle number concentrations in the 2-100 nm
range with observations for ACDC_DB_CE, which should result from a combination
of surplus newly formed particles and fast particle growth from inadequate assumptions
within the model.

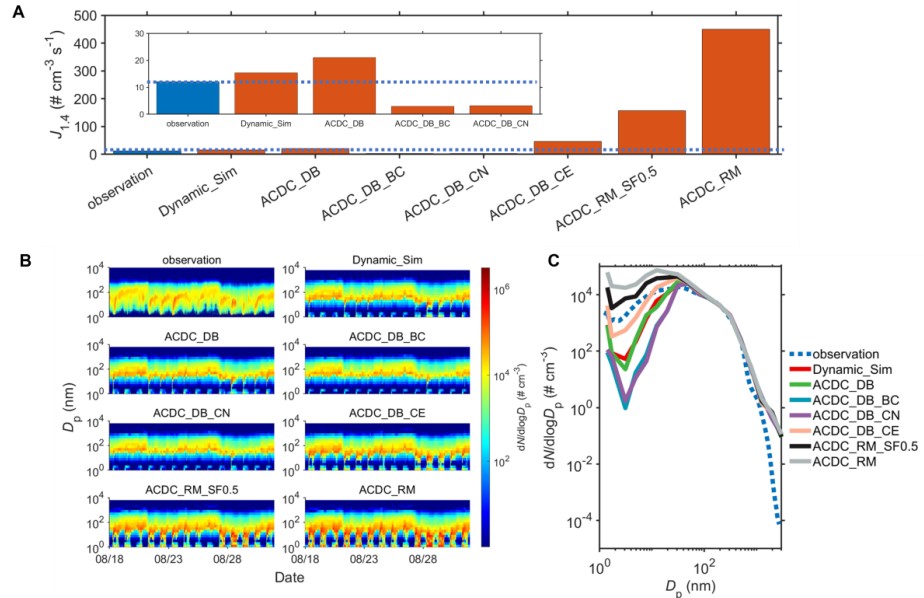

**Figure 6.** Comparison of simulated particle formation rates and particle number size distributions (PNSDs) with observations during August 18, 2019, to August 31, 2019, in Beijing. A represents the averaged particle formation rates during the period, the blue bars and orange bars represent observations and simulations, respectively, while the blue dashed line represents the observed values. Daily maximum values of $J_{1.4}$ are used following Deng et al. (2020); B for the time series of PNSDs; and C for the averaged PNSDs.

Most previous NPF studies combining experiments/observations with simulations are conducted under conditions biased towards winter (~280K) (Almeida et al., 2013; Lu et al., 2020). Under summer conditions with elevated $T$, there exists a deficiency in parameterization evaluations for simulating NPF. The 3-D simulation results during the summer period provide additional validation for the reliability of ACDC_DB. For ACDC_RM_SF0.5, evidence from both box-model simulations and 3-D simulations suggests that it can accurately reproduce real SA-DMA nucleation at temperatures around 280 K, while it has limitations in higher temperatures. Another main parameterization Dynamic_Sim consistently demonstrates good performance in NPF simulation, akin to its efficacy in winter conditions. With the increased temperature in summer (averaged $T$ is 298.2 K), the influence of simplifications in cluster evaporations, cluster number, and boundary conditions becomes more profound, mirroring the trends observed in box-model simulations above. This leads to more significant overestimation for ACDC_DB_CE, and underestimation for ACDC_DB_CN and ACDC_DB_BC compared to the observation as well as the base-case ACDC_DB. Note that CS during the summer period (averaged CS is $2.8\times10^{-2}$ s$^{-1}$) decreases compared to





winter but remains significantly higher than typical values in clean regions ($\sim3.0\times10^{-3}$
$s^{-1}$) (Dal Maso et al., 2008). According to the limited conditions for Dynamic_Sim
described above, although the overestimation of $J_{1.4}$ prediction resulting from the
simplification in cluster evaporations is more pronounced in summer compared to that
in winter, impacts from diverse overestimations and underestimations from different
simplifications and varied thermodynamics for $(SA)_1(DMA)_1$ cluster can still offset
each other, thereby allowing Dynamic_Sim to match observations. Based on previous
comparisons using box-models, significant differences in $J_{1.4}$ predictions between
Dynamic_Sim and ACDC_DB only exist under conditions of high $T > \sim300$ K and low
CS $< \sim3\times10^{-3}$ $s^{-1}$, thus similar performance of Dynamic_Sim and ACDC_DB can be
expected in the polluted atmosphere (CS $> \sim1.0\times10^{-2}$ $s^{-1}$). In clean atmosphere with
high temperature, however, caution is advised when using Dynamic_Sim for 3-D NPF
simulations.
**4. CONCLUSIONS**

By integrating box modeling, 3-D simulations, also under the constraint from in

situ measurements, this study conducts comprehensive comparison of different cluster
dynamics-based parameterizations for SA-DMA nucleation. Among them, the ACDC-
derived parameterization grounded in the latest molecular-level understanding and
complete representation of cluster dynamics (ACDC_DB), is identified to effectively
model particle formation rates and PNSDs evolution in both winter and summer in
Beijing within 3-D simulations. While a previously proposed simplified cluster
dynamics-based parameterization (Dynamic_Sim) performs comparably in modeling
NPF in Beijing, analysis reveals that their similarity arises from a delicate balance
between overestimation and underestimation due to simplifications in cluster dynamics
processes and the difference in thermodynamics of initial cluster. Particularly, under
specific conditions of high temperature ($> \sim300$ K) and low CS ($< \sim3\times10^{-3}$ $s^{-1}$),
Dynamic_Sim tends to make significant overestimation of particle formation rates
compared to the reality. Moreover, the study furnishes evidence that integrating ACDC-
derived parameterizations with the traditional theoretical approach RI-CC2/aug-cc-
pV(T+d)Z//M06-2X/6-311++G(3df,3pd) (ACDC_RM_SF0.5) effectively captures
particle formation rates and the evolution of PNSDs around 280 K, a temperature range
frequently explored in prior experiments and simulations investigating NPF (Kirkby et
al., 2011; Almeida et al., 2013; Kirkby et al., 2016; Xie et al., 2017; He et al., 2021; Ma
et al., 2019). Therefore, ACDC_RM_SF0.5 exhibits consistent applicability as other
two parameterizations at around $\sim280$ K. However, attributed to an inappropriate
representation of cluster thermodynamics, ACDC_RM_SF0.5 has limitations in
predicting particle formation rates at elevated temperatures. Overall, considering all
aspects, we recommend ACDC_DB as a more reliable parameterization for simulating
NPF across various atmospheric environments.

In addition to contributing to a more reasonable 3-D modeling of NPF, our research

further provides valuable references for the development of parameterizations for other
nucleation systems. Firstly, we demonstrate the efficacy of the DLPNO-CCSD(T)/aug-



cc-pVTZ//ωB97X-D/6-311++G(3df,3pd) level of theory in describing the
thermodynamic properties of SA-DMA clusters through comprehensive evidence. This
approach can thus be referenced when using quantum chemical calculations to obtain
thermodynamic data for other nucleation clusters, especially for other alkylamines such
as methylamine/trimethylamine-sulfuric acid clusters. It should be also noted, however,
that in some qualitative studies, e.g., comparing the enhancing potential or synergistic
effects of different precursors in SA-driven nucleation, methods other than DLPNO-
CCSD(T)/aug-cc-pVTZ//ωB97X-D/6-311++G(3df,3pd), such as RI-CC2/aug-cc-
pV(T+d)Z//M06-2X/6-311++G(3df,3pd), are equally valid (Liu et al., 2019). Secondly,
we provide comprehensive modeling evidences that certain simplifications or
assumptions in cluster dynamics, such as reducing the number of expected clusters,
modifying boundary conditions, and assuming certain clusters to be non-evaporative,
can significantly impact the prediction of particle formation rates and hence alter the 3-
D NPF simulation under certain conditions. While applying certain simplifications
concurrently under specific ambient conditions can offset different influences against
each other, leading to a satisfactory model-observation comparison, there is a risk that
certain simplifications may drive the model's outcomes away from reality when
environmental conditions change. Therefore, caution should be exercised when
applying these simplifications in derivation of nucleation parameterizations and
subsequent application in 3-D models. Lastly, we note that the development of cluster
dynamics-based nucleation parameterizations in the form of explicit mathematical
expressions is subject to limitations, especially for systems involving multiple
precursor species (Semeniuk and Dastoor, 2018). Given that the original ACDC has
been extended to involve more than two precursor species, the ACDC-derived
parameterization framework, in the form of a look-up table, is highly meaningful for
establishing parameterizations for these multi-component nucleation systems.



**Appendix.** Abbreviations used in the main text.
**SA:** sulfuric acid
**DMA:** dimethylamine
**ACDC:** Atmospheric Cluster Dynamic Code
**DB**: DLPNO-CCSD(T)/aug-cc-pVTZ//ωB97X-D/6-311++G(3df,3pd) level of theory
**RM**: RI-CC2/aug-cc-pV(T+d)Z//M06-2X/6–311++G(3df,3pd) level of theory
**CE**: simplification in cluster evaporations (only $(SA)_k(DMA)_k$ ($k$ = 1-4) and
$(SA)_2(DMA)_1$ clusters are considered)
**CN**: simplification in cluster number (clusters larger than $(SA)_1(DMA)_1$ are regarded
stable with no evaporation)
**BC**: simplification in boundary conditions ($(SA)_4(DMA)_4$ cluster is set as the only
terminal cluster in calculating particle formation rates)
**SF**: sticking factor used in collision process
**Dynamic_Sim:** a reported cluster-dynamic based parameterization incorporating
simplifications of CE, CN and BC.
$J_{1.4}$: particle formation rate at 1.4 nm
$R$: a parameter to quantify the differences in simulating $J_{1.4}$ among different cluster
dynamics-based parameterizations compared to the base-case ACDC_DB



**Code and data availability.** The data and code used in this study are available upon
request from the corresponding author.
**Author contributions**. JS, BZ, and SW designed the research; AN and XZ collected
the quantum chemistry calculation data; JS performed the ACDC and WRF-
Chem/R2D-VBS simulations; YL, RC, and JJ collected the observational data. JS, BZ,
and SW analyzed the data; RC, DG, JJ, YG, MS, BC, and HH presented important
suggestions for the paper; JS, BZ, and SW wrote the paper with input from all co-
authors.
**Competing interests.** At least one of the (co-)authors is a member of the editorial board
of Atmospheric Chemistry and Physics.

**Acknowledgements.** This study was supported by the National Natural Science
Foundation of China (22188102 and 42275110).
**Financial support.** Financial support from National Natural Science Foundation of
China (22188102 and 42275110).



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
