# Peer review of "Cluster Dynamics-based Parameterization for Sulfuric Acid-Dimethylamine"

_EGUsphere, 2024_

## Referee Comment (RC1)

**General comments:**

The work conducts comprehensive comparison of different cluster dynamics-based parameterizations for SA-DMA nucleation by integrating box-model simulations, 3-D modeling, and in-situ observations. It is found that ACDC_DB performs well in modeling 3-D NPF for both winter and summer in Beijing and shows promise for application in various atmospheric environments. Furthermore, ACDC_RM_SF0.5 exhibits effective applicability at ~280 K, but has limitations in predicting $J_{1.4}$ at elevated T. While Dynamic_Sim is applicable for simulating NPF in polluted atmospheres but makes significant overestimation of $J_{1.4}$ under conditions of high T and low CS.

The topic discussed in this paper is highly meaningful for developing parameterizations for various nucleation systems. The reported results are clearly presented and are relevant to the scope of *Atmos. Chem. Phys.* I recommend publication of this manuscript after consideration of the following comments.

**Specific comments:**

1) Lines 94-95: Please explain briefly the reason for considering such three simplifications within Dynamics_Sim.

2) Line 281: To make a clear understanding among readers, it would be better to provide the concept of the chemical initial and boundary conditions in WRF-Chem/R2D-VBS simulations.

3) Figure 6C: It can be noted that ACDC_DB and Dynamic_Sim also exhibit an underestimation of averaged PNSDs in the 2-100 nm range in comparison to observation. Can the authors account for the cause of this phenomenon?

**Technical corrections:**

1) Lines 143-144: "*n* and *m* represent the number of SA and DMA molecules in a cluster" should be "*m* and *n* represent the number of SA and DMA molecules in a cluster".

2) Lines 465 and 482: "ACDC_RM" should be "ACDC_RM_SF0.5".

3) Supporting Information, lines 37 and 38: "A: $\Delta G$ = 13.5 kcal/mol; B: $\Delta G$ = 12.9 kcal/mol" should be "A: $\Delta G$ = -13.5 kcal/mol; B: $\Delta G$ = -12.9 kcal/mol".

---

## Community Comment (CC1)

The manuscript by Shen et al. presents simulations of new particle formation from sulfuric acid ($H_2SO_4$) and dimethylamine (DMA) by molecular cluster kinetics modeling. The thorough comparisons of formation rates obtained with different thermochemistry input data sets and kinetic model assumptions provide very useful information on variations and uncertainties in predicted formation rates.

I would like to bring up previous works applying ACDC-based particle formation rate look-up tables in large-scale 3D modeling, as the authors may not be aware of them (e.g. L118-119). These previous studies have applied look-up tables in the PMCAMx-UF, GEOS-Chem-TOMAS and EC-Earth3 chemical transport or Earth system models with the following particle formation mechanisms:

- $H_2SO_4$–$NH_3$–$H_2O$ with electrically neutral clusters (Baranizadeh et al., 2016; Croft et al., 2016),
- $H_2SO_4$–$NH_3$–$H_2O$ + $H_2SO_4$–DMA with neutral clusters (Julin et al., 2018; Olin et al., 2022), and
- $H_2SO_4$–$NH_3$ including both neutral and ionic species (Svenhag et al., 2024).

Most of these studies applied quantum chemistry data corresponding to the RICC2 method, as those were the only available complete data sets at the time. A comparison of global $H_2SO_4$–$NH_3$ particle formation and its effects as predicted by either DLPNO or RICC2 is presented by Svenhag et al. (2024). While DLPNO is the current best available method, it may underpredict formation rates under certain conditions (e.g. Besel et al., 2020), and the DLPNO-based rates were thus applied to assess the lower limits of the predicted effects.

In the first studies, the calculation and interpolation of look-up tables were hard-coded for the given chemical components. The most recent work (Svenhag et al., 2024) applies automatized look-up table generator and interpolator that are applicable to arbitrary components (Yazgi and Olenius, 2023), enabling easy incorporation of tables obtained for different species and thermochemistry data. Automatization is needed especially for reading in and interpolating tables within the 3D model, as it is not feasible to maintain separate interpolation routines for different tables, corresponding to different chemical mechanisms and/or dimensions.

It can also be noted that the usage of pre-calculated formation rates (which is necessary in computationally heavy 3D models) involves simplifying assumptions on gas–particle kinetics, as there are no explicit interactions between the clusters and the nucleating vapors and larger nanoparticles. Therefore, a parameterization or look-up table approach may give biased results under some conditions even if the thermochemistry data were perfectly accurate. In computationally light-weight models, this can be circumvented by explicit simulation of the coupled gas–cluster–aerosol system (Olenius and Roldin, 2022), corresponding to a multi-component adaptation of discrete–sectional modeling (Li and Cai, 2020).

Finally, I also encourage to refer to the ACDC code repository (https://github.com/tolenius/ACDC) in order to provide a reference for the model tools for reproducibility of simulation results.

**References**

Baranizadeh, E. et al.: Implementation of state-of-the-art ternary new-particle formation scheme to the regional chemical transport model PMCAMx-UF in Europe. *Geosci. Model Dev.*, 9, 2741–2754, https://doi.org/10.5194/gmd-9-2741-2016, 2016

Besel, V. et al: Impact of quantum chemistry parameter choices and cluster distribution model settings on modeled atmospheric particle formation rates. *J. Phys. Chem. A*, 124, 5931−5943, https://dx.doi.org/10.1021/acs.jpca.0c03984, 2020

Croft, B. et al.: Contribution of Arctic seabird-colony ammonia to atmospheric particles and cloud-albedo radiative effect. *Nat. Commun.* 7, 13444, https://doi.org/10.1038/ncomms13444, 2016

Julin, J. et al.: Impacts of future European emission reductions on aerosol particle number concentrations accounting for effects of ammonia, amines, and organic species. *Environ. Sci. Technol.,* 52, 692-700, https://doi.org/10.1021/acs.est.7b05122, 2018

Li, C. and Cai, R.: Tutorial: The discrete-sectional method to simulate an evolving aerosol. *J. Aerosol Sci.*, 150, 105615, https://doi.org/10.1016/j.jaerosci.2020.105615, 2020

Olenius, T. and Roldin, P.: Role of gas–molecular cluster–aerosol dynamics in atmospheric new-particle formation. *Sci. Rep.*, 12, 10135, https://doi.org/10.1038/s41598-022-14525-y, 2022

Olin, M. et al.: Contribution of traffic-originated nanoparticle emissions to regional and local aerosol levels. *Atmos. Chem. Phys.*, 22, 1131–1148, https://doi.org/10.5194/acp-22-1131-2022, 2022

Svenhag, C. et al.: Implementing detailed nucleation predictions in the Earth system model EC-Earth3.3.4: sulfuric acid-ammonia nucleation. *EGUsphere [preprint]*, https://doi.org/10.5194/egusphere-2023-2665, 2024

Yazgi and Olenius: J-GAIN v1.1: a flexible tool to incorporate aerosol formation rates obtained by molecular models into large-scale models. *Geosci. Model Dev.*, 16, 5237–5249, https://doi.org/10.5194/gmd-16-5237-2023, 2023

---

## Author Comment (AC1)

**Reviewer #1**

This manuscript compared the simulations of new particle formation rate from sulfuric acid ($H_2SO_4$) and dimethylamine (DMA) by different molecular cluster kinetics modeling under various conditions (e.g., different T and CS) and aims to find applicable parameterizations for 3-D NPF modeling in Beijing. The provided results indicated that: ACDC_DB, an ACDC derived parameterization, incorporated into WRF-Chem/R2D-VBS model can effectively reproduce particle formation rates and PNSDs evolution for both winter and summer in Beijing. The content of this paper is useful for developing parameterizations aiming at predicting or simulating NPF in urban areas. The manuscript is well written and the topic fits the scope of Atmos. Chem. Phys. I recommend publication of this manuscript after responding the following comments.

**Response:** We sincerely thank for the reviewer's careful review of our manuscript and the positive comments.

Specific comments:

The authors should clarify the advantage or difference of the parameterizations developed using ACDC to the one used in Dynamic_Sim. On the words, why the authors developed ACDC based parameterizations rather than making iteration on the base of Dynamic_Sim in L173-176 and in the introduction part. Moreover, except for making comparison with Dynamic_Sim, what's the consideration of setting the ACDC_BC coupling with three simplifications?

**Response:** In 3-D modeling, calculating detailed real-time nucleation dynamics for arbitrary chemical systems can impose a substantial computational burden, presenting a significant challenge for direct implementation (Yazgi and Olenius, 2023). Practical approaches for integrating NPF mechanisms into 3-D models generally follow two strategies: 1) simplifying the processes to derive explicit mathematical expressions, such as Dynamic_Sim, or 2) utilizing precomputed look-up tables generated from other box-model simulations, such as ACDC-derived parameterizations. Therefore, the primary difference between Dynamic_Sim and ACDC-derived parameterizations lies in the manner of their integration into the 3-D model.

Beyond that, a key difference between the main ACDC-derived parameterization ACDC_DB and Dynamic_Sim lies in their ability to accurately represent cluster dynamics and, consequently, particle formation rates under different atmospheric conditions. Our results suggest that simplifying the cluster dynamic processes in Dynamic_Sim may introduce biases compared to the comprehensive treatment (Figure 2), resulting in an overestimation for Dynamic_Sim in $J_{1.4}$ prediction relative to ACDC_DB under high temperature (> ~300 K) and low CS conditions (< ~$3\times10^{-3}$ s$^{-1}$) (Figure 1).

Based on our previous study (Li et al., 2023), removing the three inherent assumptions in Dynamic_Sim would escalate the computational demand by over several orders of magnitude. Consequently, we chose the alternative approach of employing ACDC-derived look-up tables for probing the impacts of simplifications and subsequent comparison and selection of SA-DMA nucleation parameterization. We have added relevant explanations to line 191-195 in the revised manuscript. The

purpose of establishing an ACDC-derived parameterization that simultaneously couples three simplifications is to elucidate the differences between ACDC_DB and Dynamic_Sim. Our results (Figure S1) indicate that the discrepancies between the two arise from only the three simplifications in cluster dynamics and the thermodynamics of the initial cluster, while other dynamic processes remain consistent.

The look-up table approach has its limitation due to the ignorance of the explicit interactions of clusters with gas phase precursors and pre-existing particles. The author should add some discussion about the disadvantage of the applied look-up table approach and discuss about the possible conditions that may lead to the biased simulation results.

**Response:** We appreciate the reviewer's suggestions, which can help to improve the quality of our manuscript. We agree that the explicit interactions among gaseous precursors, clusters, and pre-existing particles are simplified when using a look-up table approach, though it is commonly used in 3-D modeling. Additional analysis and discussion on the potential impacts of this simplification have been added to the *CONCLUSIONS and DISCUSSIONS* sections. Olenius and Roldin (2022) provided insights on the potential impact of gas–cluster–aerosol dynamics on NPF simulation using chemical transport models. Among various standard treatments of gas–cluster–aerosol dynamics in chemical transport modeling, they highlighted the assumption of instantaneous steady-state nucleation at every model time step as a potential source of bias. In response to this, we conducted a reliability assessment of steady-state nucleation in our WRF-Chem/R2D-VBS simulations. We evaluated the validity of the steady-state nucleation assumption by considering the system's e-folding time (time for clusters to reach (1-1/e) of their terminal concentration, following Li et al., (2023)). Specifically, we deemed the assumption reasonable if, under certain atmospheric conditions, the system's e-folding time is less than the simulation time step (300 s).

As shown in Figure S14, results indicates that the e-folding time does not show a significant correlation with $J_{1.4}$. Under the majority of atmospheric conditions (77.3%), the nucleating system's e-folding time is less than 300 s. Instances where the e-folding time exceeds 300 s are primarily observed in winter clean conditions characterized by low temperature (T < ~270 K), low condensation sink (CS < ~0.003 s$^{-1}$), and low precursor concentrations (SA < ~$10^6$ cm$^{-3}$). These findings align with the observations of Olenius and Roldin (2022). It's important to emphasize that this e-folding time represents the duration required for the system to transition from having only precursor molecules to reaching near-equilibrium concentrations of various clusters. In reality, cluster concentrations generally do not start from zero. Therefore, the calculated e-folding time serves as an upper limit estimate. Given the predominance of atmospheric conditions where the e-folding time falls within or below the simulation time step of 300 s, consequently, the steady-state treatment is generally deemed reasonable for our WRF-Chem/R2D-VBS simulations.

[Figure]

**Figure S14.** The variation of e-folding time with $J_{1.4}$ correlated with temperature (A), CS (B), SA concentration (C), and DMA concentration (D). The data points were calculated using a more sparse sequence of input parameters (T: 250, 260, 270, 280, 290, 300, 310, 320 (K); CS: $5.00 \times 10^{-4}$, $5.00 \times 10^{-3}$, $5.00 \times 10^{-2}$, $5.00 \times 10^{-1}$ (s$^{-1}$); SA: $1.00 \times 10^{5}$, $1.00 \times 10^{6}$, $1.00 \times 10^{7}$, $1.00 \times 10^{8}$ (cm$^{-3}$); DMA: $5.00 \times 10^{6}$, $5.00 \times 10^{7}$, $5.00 \times 10^{8}$ (cm$^{-3}$)) compared to those shown in Table S1.

We further investigated another common treatment that may introduce bias: neglecting cluster formation in consuming precursor during nucleation. Our examination focused on assessing the proportion of precursor consumption by cluster formation relative to precursor concentrations. As shown in Figure S15 and S16, we found that this proportion increases with $J_{1.4}$ for both SA and DMA. Under the majority of atmospheric conditions (82.0% for DMA and 57% for SA), proportions are below 10%. Proportions exceed 10% are predominantly observed in scenarios also characterized by low temperature (T < ~270 K) and low condensation sink (CS < ~0.003 s$^{-1}$), but with high deference in concentrations between DMA and SA. Specifically, elevated SA concentrations, which lead to significant DMA consumption through cluster formation, and vice versa, contribute to scenarios where precursor consumption by cluster formation exceeds 10%. It's noteworthy that our calculation of precursor consumption by cluster formation starts from zero cluster concentration. Also, in the real atmosphere, cluster concentrations are generally nonzero, leading to another upper limit estimate. Therefore, based on our analysis, it can be inferred that cluster formation may not introduce significant bias into NPF simulations under typical atmospheric conditions. We have added this additional analysis and discussion of the potential impacts of these common treatments in NPF simulations to line 717-731 in the revised manuscript and the supporting information.

[Figure]

Figure S15. The variation of proportion of DMA consumption by cluster formation relative to precursor concentrations with $J_{1.4}$, correlated with temperature (A), CS (B), SA concentration (C), and DMA concentration (D). The input variables are consistent with Figure S14.

[Figure]

Figure S16. The variation of proportion of SA consumption by cluster formation relative to precursor concentrations with $J_{1.4}$, correlated with temperature (A), CS (B), SA concentration (C), and DMA concentration (D). The input variables are consistent with Figure S14.

In Figure S8, it seems to me that ACDC_RM_SF0.5 overestimate the formation rate by a factor of 2 at 293K, please check the simulation results or discuss the possible reasons. Would this influence the 3D model simulations during summer, leading to the overestimation of $J_{1.4}$? Moreover, Figure S8 also indicated that ACDC_DB and

Dynamic_Sim overestimate the formation rate more at 293K compared with at 278K, would this be the reason of the overestimation during summer?

**Response:** Figure S8 compares box-model simulations from three main parameterizations, ACDC_DB, Dynamic_Sim, and ACDC_RM_SF0.5, with the well-controlled CLOUD chamber experiments. The results reveal that under the conditions of 278 K, all three parameterizations are consistent with the CLOUD chamber experiments. This alignment mirrors our 3-D simulation for winter Beijing (Figure 5A), which also corresponds to a similar temperature (~274.7 K). However, at 293 K, while ACDC_DB and Dynamic_Sim remain close to the observations, ACDC_RM_SF0.5 substantially overestimates the particle formation rate by more than an order of magnitude. Additionally, our summer simulation for Beijing, illustrated in Figure 6A, demonstrates that ACDC_RM_SF0.5 significantly overestimates particle formation rates compared to those derived from in situ observations at ~298.2 K. Hence, the patterns shown in Figure S8, Figure 5A, and Figure 6A are actually consistent. The inability of ACDC_RM_SF0.5 to accurately simulate particle formation rates at high temperatures can be attributed to its inappropriate representation of cluster thermodynamics as explained in line 468-491 in the revised manuscript.

In Figure S8, we used the average DMA concentration for the box-model simulation, whereas the DMA concentration for each data point from Xiao et al. (2021) might differ slightly. Here, we re-simulated these cases using ACDC_DB with the precursor concentrations corresponding to each particle formation rate from Xiao et al. (2021). As shown in Figure S9A, the simulations at both 278 K and 293 K generally align with the experimental values from Xiao et al. (2021) within a factor of two. The box-model simulations at 293 K tend to slightly overestimate the particle formation rates. This discrepancy may arise because the CLOUD chamber measured particles with a diameter of 1.7 nm, while our simulations modeled the formation of particles with a diameter of 1.4 nm, which may be slightly higher (Almeida et al. 2013). According to the modified Kerminen-Kulmala equation (Lehtinen et al., 2007), the difference between the formation rates of 1.4-nm and 1.7-nm particles should be related to the growth rate of clusters. In the SA-DMA nucleation system, SA-DMA clusters with different molecular ratios are the main materials for growth. We further compared the differences in SA-DMA cluster concentrations at two temperatures. As shown in Figure S9B, $(SA)_1(DMA)_1$ has the highest concentration among all SA-DMA clusters and is likely the most critical cluster contributing to growth, consistent with previous studies (Almeida et al. 2013; Cai et al. 2023). Notably, the concentration of $(SA)_1(DMA)_1$ cluster at 278 K is about an order of magnitude higher than that at 293 K. Therefore, this will result in the particle formation rate of 1.4-nm particles at 278 K being closer to the particle formation rate of 1.7-nm particles in the CLOUD chamber compared to that at 293 K. In Figure 6A, the particle formation rate derived from the 3-D simulation using ACDC_DB is higher than the observed rate, likely due to the slight overestimation of the SA concentration during this period (Table S2). We have added these additional analysis and discussion to line 511-515 in the revised manuscript.

[Figure]

Figure S9. Comparison of measured $J_{1.7}$ from Xiao et al. 2021 and simulated $J_{1.4}$ using ACDC_DB with corresponding DMA concentrations in experiments (A), and the comparison of cluster concentrations at 293 K and 278 K (B).

Technical comments:
L90-91: check the reference

**Response:** The revisions have been made accordingly.

Lines 465 and 482: "ACDC_RM" should be "ACDC_RM_SF0.5"
**Response:** The revisions have been made accordingly.

Line 476-478 and other parts in section 3.1: I suggest using "overestimate" and "underestimate" instead of "applicable "and "suitable ", since the discussion in section 3.1 is the evaluation of different simplifications on the molded J1.4 for ADCD_RM and ACDC_DB.
**Response:** We agree with the reviewer that "overestimate" and "underestimate" is better. The revisions have been made accordingly.

Line 525: Replace the comma of 'ACDC_RM show higher concentrations,' with period.
**Response:** The revisions have been made accordingly.

**REFERENCES**

Almeida, J., Schobesberger, S., Kurten, A., Ortega, I. K., Kupiainen-Maatta, O., Praplan, A. P., Adamov, A., Amorim, A., Bianchi, F., Breitenlechner, M., David, A., Dommen, J., Donahue, N. M., Downard, A., Dunne, E., Duplissy, J., Ehrhart, S., Flagan, R. C., Franchin, A., Guida, R., Hakala, J., Hansel, A., Heinritzi, M., Henschel, H., Jokinen, T., Junninen, H., Kajos, M., Kangasluoma, J., Keskinen, H., Kupc, A., Kurten, T., Kvashin, A. N., Laaksonen, A., Lehtipalo, K., Leiminger, M., Leppa, J., Loukonen, V., Makhmutov, V., Mathot, S., McGrath, M. J., Nieminen, T., Olenius, T., Onnela, A., Petaja, T., Riccobono, F., Riipinen, I., Rissanen, M., Rondo, L., Ruuskanen, T., Santos, F. D., Sarnela, N., Schallhart, S., Schnitzhofer, R., Seinfeld, J. H., Simon, M., Sipila, M., Stozhkov, Y., Stratmann, F., Tome, A., Trostl, J., Tsagkogeorgas, G., Vaattovaara, P., Viisanen, Y., Virtanen, A., Vrtala, A., Wagner, P. E., Weingartner, E., Wex, H., Williamson, C., Wimmer, D., Ye, P., Yli-Juuti, T., Carslaw, K. S., Kulmala, M., Curtius, J., Baltensperger, U., Worsnop, D. R., Vehkamaki, H., and Kirkby, J.: Molecular understanding of sulphuric acid-amine particle nucleatio

Cai, R., Yin, R., Li, X., Xie, H.-B., Yang, D., Kerminen, V.-M., Smith, J. N., Ma, Y., Hao, J., Chen, J., Kulmala, M., Zheng, J., Jiang, J., and Elm, J.: Significant contributions of trimethylamine to sulfuric acid nucleation in polluted environments, npj Climate and Atmospheric Science, 6, 10.1038/s41612-023-00405-3, 2023.

Lehtinen, K. E. J., Dal Maso, M., Kulmala, M., and Kerminen, V. M.: Estimating nucleation rates from apparent particle formation rates and vice versa: Revised formulation of the Kerminen-Kulmala equation, Journal of Aerosol Science, 38, 988-994, 10.1016/j.jaerosci.2007.06.009, 2007.

Li, Y., Shen, J., Zhao, B., Cai, R., Wang, S., Gao, Y., Shrivastava, M., Gao, D., Zheng, J., Kulmala, M., and Jiang, J.: A dynamic parameterization of sulfuric acid–dimethylamine nucleation and its application in three-dimensional modeling, Atmospheric Chemistry and Physics, 23, 8789-8804, 10.5194/acp-23-8789-2023, 2023.

Olenius, T. and Roldin, P.: Role of gas-molecular cluster-aerosol dynamics in atmospheric new-particle formation, Sci Rep, 12, 10135, 10.1038/s41598-022-14525-y, 2022.

Xiao, M., Hoyle, C. R., Dada, L., Stolzenburg, D., Kürten, A., Wang, M., Lamkaddam, H., Garmash, O., Mentler, B., Molteni, U., Baccarini, A., Simon, M., He, X.-C., Lehtipalo, K., Ahonen, L. R., Baalbaki, R., Bauer, P. S., Beck, L., Bell, D., Bianchi, F., Brilke, S., Chen, D., Chiu, R., Dias, A., Duplissy, J., Finkenzeller, H., Gordon, H., Hofbauer, V., Kim, C., Koenig, T. K., Lampilahti, J., Lee, C. P., Li, Z., Mai, H., Makhmutov, V., Manninen, H. E., Marten, R., Mathot, S., Mauldin, R. L., Nie, W., Onnela, A., Partoll, E., Petäjä, T., Pfeifer, J., Pospisilova, V., Quéléver, L. L. J., Rissanen, M., Schobesberger, S., Schuchmann, S., Stozhkov, Y., Tauber, C., Tham, Y. J., Tomé, A., Vazquez-

Pufleau, M., Wagner, A. C., Wagner, R., Wang, Y., Weitz, L., Wimmer, D., Wu, Y., Yan, C., Ye, P., Ye, Q., Zha, Q., Zhou, X., Amorim, A., Carslaw, K., Curtius, J., Hansel, A., Volkamer, R., Winkler, P. M., Flagan, R. C., Kulmala, M., Worsnop, D. R., Kirkby, J., Donahue, N. M., Baltensperger, U., El Haddad, I., and Dommen, J.: The driving factors of new particle formation and growth in the polluted boundary layer, Atmospheric Chemistry and Physics, 21, 14275-14291, 10.5194/acp-21-14275-2021, 2021.

Yazgi, D. and Olenius, T.: J-GAIN v1.1: a flexible tool to incorporate aerosol formation rates obtained by molecular models into large-scale models, Geoscientific Model Development, 16, 5237-5249, 10.5194/gmd-16-5237-2023, 2023.

---

## Author Comment (AC2)

**Reviewer #2**

General comments:

The work conducts comprehensive comparison of different cluster dynamics-based parameterizations for SA-DMA nucleation by integrating box-model simulations, 3-D modeling, and in-situ observations. It is found that ACDC_DB performs well in modeling 3-D NPF for both winter and summer in Beijing and shows promise for application in various atmospheric environments. Furthermore, ACDC_RM_SF0.5 exhibits effective applicability at ~280 K, but has limitations in predicting $J_{1.4}$ at elevated T. While Dynamic_Sim is applicable for simulating NPF in polluted atmospheres but makes significant overestimation of $J_{1.4}$ under conditions of high T and low CS.

The topic discussed in this paper is highly meaningful for developing parameterizations for various nucleation systems. The reported results are clearly presented and are relevant to the scope of Atmos. Chem. Phys. I recommend publication of this manuscript after consideration of the following comments.

**Response:** We thank the reviewer for recognizing and recommending our work.

Specific comments:

1) Lines 94-95: Please explain briefly the reason for considering such three simplifications within Dynamics_Sim.

**Response:** Generally, according to theoretical studies (Olenius et al., 2013, 2017; Ortega et al., 2012; Myllys et al., 2019), clusters $(SA)_1(DMA)_1$, $(SA)_1(DMA)_2$, $(SA)_2(DMA)_2$, $(SA)_3(DMA)_3$ and $(SA)_4(DMA)_4$ are considered the key clusters along the cluster formation pathways in SA-DMA nucleation. Under the polluted conditions (CS > ~$1.0 \times 10^{-2}$ s$^{-1}$), the evaporation rates of clusters $(SA)_1(DMA)_2$, $(SA)_2(DMA)_2$, $(SA)_3(DMA)_3$ and $(SA)_4(DMA)_4$ are negligible compared to their coagulation sink. Therefore, three simplifications are involved in derivation of Dynamic_Sim as described in line 93-102 in the revised manuscript. Details of the derivation of Dynamic_Sim can be seen in our previous study (Li et al. 2023).

2) Line 281: To make a clear understanding among readers, it would be better to provide the concept of the chemical initial and boundary conditions in WRF-Chem/R2D-VBS simulations.

**Response:** We appreciate the reviewer's suggestion which can help to enhance the readability of our manuscript. The chemical initial condition in WRF-Chem/R2D-VBS simulations refers to the concentration field of gas-phase/particulate chemical variables at the beginning of the simulation, standing as the evolution of these species before the simulation duration. The chemical boundary condition here refers to the fluxes or concentrations at the edges of the simulated domain (Brasseur et al. 2017). In WRF-Chem/R2D-VBS simulations, we use a 5-day spin-up to minimize the impact of chemical initial conditions on simulation results. Some explanations have been added in line 296-307 in the revised manuscript.

3) Figure 6C: It can be noted that ACDC_DB and Dynamic_Sim also exhibit an

underestimation of averaged PNSDs in the 2-100 nm range in comparison to observation. Can the authors account for the cause of this phenomenon?

**Response:** In fact, ACDC_DB and Dynamic_Sim do not exhibit a consistent underestimation of averaged PNSDs along the 2-100 nm range in comparison to observation. Similar to wintertime simulation, the PNSDs simulated by ACDC_DB and Dynamic_Sim for the summer season are relatively overestimated compared to observations in larger size range of 30-100 nm. This may not be evident due to overlapping curves in Figure 6C of the main text but is more noticeable in Figure A1 below.

[Figure]

Figure A1. Comparison of observed and simulated PNSDs during August 18, 2019, to August 31, 2019, in Beijing. Simulations are conducted using parameterizations of Dynamic_Sim, ACDC_DB, ACDC_DB_CE, and ACDC_RM_SF0.5.

For the 2-100 nm range, we also compared the total number concentrations simulated from three main parameterizations and the ACDC_DB_CE with the observations (Figure S13). It can be noted that the number concentrations simulated by ACDC_DB and Dynamic_Sim are relatively consistent with the observations, whereas ACDC_DB_CE and another main parameterization ACDC_RM_SF0.5 tend to overestimate the number concentrations by a factor of 1.6 and 2.5, respectively. Combining the particle formation rates shown in Figure 6A for the three parameterizations, it can be concluded that the total concentrations of 2-100 nm particles are primarily influenced by nucleation. The discrepancies in PNSDs across different size ranges compared to the observations arise from the intrinsic treatment of growth processes in the 3-D model. We have added relevant clarifications in line 620-627 in the revised manuscript.

[Figure]

Figure S13. Comparison of observed and simulated aerosol number concentration within 2-100 nm during August 18, 2019, to August 31, 2019, in Beijing. Simulations are conducted using parameterizations of Dynamic_Sim, ACDC_DB, ACDC_DB_CE, and ACDC_RM_SF0.5.

Technical corrections:
1) Lines 143-144: "n and m represent the number of SA and DMA molecules in a cluster" should be "m and n represent the number of SA and DMA molecules in a cluster".
**Response:** The revisions have been made accordingly.

2) Lines 465 and 482: "ACDC_RM" should be "ACDC_RM_SF0.5".
**Response:** The revisions have been made accordingly.

3) Supporting Information, lines 37 and 38: "A: $\Delta G$ = 13.5 kcal/mol; B: $\Delta G$ = 12.9 kcal/mol" should be "A: $\Delta G$ = -13.5 kcal/mol; B: $\Delta G$ = -12.9 kcal/mol".
**Response:** The revisions have been made accordingly.

**REFERENCES**

Brasseur GP, Jacob DJ. Model Equations and Numerical Approaches. In: Modeling of Atmospheric Chemistry. Cambridge University Press; 2017:84-204.

Li, Y., Shen, J., Zhao, B., Cai, R., Wang, S., Gao, Y., Shrivastava, M., Gao, D., Zheng, J., Kulmala, M., and Jiang, J.: A dynamic parameterization of sulfuric acid–dimethylamine nucleation and its application in three-dimensional modeling, Atmospheric Chemistry and Physics, 23, 8789-8804, 10.5194/acp-23-8789-2023, 2023.

Myllys, N., Chee, S., Olenius, T., Lawler, M., and Smith, J.: Molecular-Level Understanding of Synergistic Effects in Sulfuric Acid-Amine-Ammonia Mixed Clusters, J Phys Chem A, 123, 2420-2425, 10.1021/acs.jpca.9b00909, 2019.

Olenius, T., Kupiainen-Maatta, O., Ortega, I. K., Kurten, T., and Vehkamaki, H.: Free energy barrier in the growth of sulfuric acid-ammonia and sulfuric acid-dimethylamine clusters, J Chem Phys, 139, 084312, 10.1063/1.4819024, 2013.

Olenius, T., Halonen, R., Kurtén, T., Henschel, H., Kupiainen‐Määttä, O., Ortega, I. K., Jen, C. N., Vehkamäki, H., and Riipinen, I.: New particle formation from sulfuric acid and amines: Comparison of monomethylamine, dimethylamine, and trimethylamine, Journal of Geophysical Research: Atmospheres, 122, 7103-7118, 10.1002/2017jd026501, 2017.

Olenius, T. and Roldin, P.: Role of gas-molecular cluster-aerosol dynamics in atmospheric new-particle formation, Sci Rep, 12, 10135, 10.1038/s41598-022-14525-y, 2022.

Ortega, I. K., Kupiainen, O., Kurtén, T., Olenius, T., Wilkman, O., McGrath, M. J., Loukonen, V., and Vehkamäki, H.: From quantum chemical formation free energies to evaporation rates, Atmospheric Chemistry and Physics, 12, 225-235, 10.5194/acp-12-225-2012, 2012.

---

## Author Comment (AC3)

**Community Comment:**

The manuscript by Shen et al. presents simulations of new particle formation from sulfuric acid ($H_2SO_4$) and dimethylamine (DMA) by molecular cluster kinetics modeling. The thorough comparisons of formation rates obtained with different thermochemistry input data sets and kinetic model assumptions provide very useful information on variations and uncertainties in predicted formation rates.

**Response:** We highly appreciate Dr. Olenius's attention to our study. We have carefully studied the comments, which are very helpful in improving the quality of our manuscript. We have provided detailed responses to each point of comment and made revisions in the manuscript accordingly.

I would like to bring up previous works applying ACDC-based particle formation rate look-up tables in large-scale 3D modeling, as the authors may not be aware of them (e.g. L118-119). These previous studies have applied look-up tables in the PMCAMx-UF, GEOS-ChemTOMAS and EC-Earth3 chemical transport or Earth system models with the following particle formation mechanisms:

• $H_2SO_4$–$NH_3$–$H_2O$ with electrically neutral clusters (Baranizadeh et al., 2016; Croft et al., 2016),

• $H_2SO_4$–$NH_3$–$H_2O$ + $H_2SO_4$–DMA with neutral clusters (Julin et al., 2018; Olin et al., 2022),

and

• $H_2SO_4$–$NH_3$ including both neutral and ionic species (Svenhag et al., 2024).

**Response:** We thank Dr. Olenius for providing these research summaries. We have revised the relevant sections in the revised manuscript to discuss and review the previous studies applying ACDC-based particle formation rate look-up tables in 3-D modeling in line 122-132 in the revised manuscript.

Most of these studies applied quantum chemistry data corresponding to the RICC2 method, as those were the only available complete data sets at the time. A comparison of global $H_2SO_4$–$NH_3$ particle formation and its effects as predicted by either DLPNO or RICC2 is presented by Svenhag et al. (2024). While DLPNO is the current best available method, it may underpredict formation rates under certain conditions (e.g. Besel et al., 2020), and the DLPNO-based rates were thus applied to assess the lower limits of the predicted effects.

**Response:** As mentioned above, although some studies have used ACDC-derived look-up tables in 3-D models for NPF simulations, the impact of input thermodynamic data, especially RI-CC2 and DLPNO on 3-D NPF simulation involving SA-DMA nucleation, is not yet clear. We have clarified the specific research gap addressed by our study in lines 132-138 in the revised manuscript.

We also agree that DLPNO may have uncertainties in fully accurately describing cluster thermodynamics, despite being currently recognized as the best quantum chemical calculation methods. We have clarified this in line 700-702 in the revised manuscript.

In the first studies, the calculation and interpolation of look-up tables were hard-coded for the given chemical components. The most recent work (Svenhag et al., 2024) applies automatized look-up table generator and interpolator that are applicable to arbitrary components (Yazgi and Olenius, 2023), enabling easy incorporation of tables obtained for different species and thermochemistry data. Automatization is needed especially for reading in and interpolating tables within the 3D model, as it is not feasible to maintain separate interpolation routines for different tables, corresponding to different chemical mechanisms and/or dimensions.

**Response:** We greatly appreciate the most recent works mentioned, which facilitate the integration of ACDC-based cluster dynamic simulations with 3-D modeling. For the single SA-DMA nucleation system focused in our study, the use of a hard-coded method is acceptable. However, if multiple nucleation mechanisms with different dimensions are simulated through look-up tables, the hard-coded method should be redundant. In such cases, the novel method of an automatized look-up table generator and interpolator would be much more feasible. We have added discussions on this topic in line 738-741 in the revised manuscript.

It can also be noted that the usage of pre-calculated formation rates (which is necessary in computationally heavy 3D models) involves simplifying assumptions on gas–particle kinetics, as there are no explicit interactions between the clusters and the nucleating vapors and larger nanoparticles. Therefore, a parameterization or look-up table approach may give biased results under some conditions even if the thermochemistry data were perfectly accurate. In computationally light-weight models, this can be circumvented by explicit simulation of the coupled gas–cluster–aerosol system (Olenius and Roldin, 2022), corresponding to a multicomponent adaptation of discrete–sectional modeling (Li and Cai, 2020).

**Response:** The potential impact of simplification of gas–particle kinetics using pre-calculated formation rates has also been concerned by reviewer #1. We have carefully studied the research conducted by Olenius and Roldin (2022) which represents the primary study concerning the explicit simulation of the coupled gas–cluster–aerosol system. We have examined the key dynamic processes, on which the usage of standard approach might exerts significant bias as demonstrated in this study.

Firstly, we evaluated the validity of the steady-state nucleation assumption by considering the system's e-folding time (time for clusters to reach (1-1/e) of their terminal concentration, following Li et al., (2023)). Specifically, we deemed the assumption reasonable if, under certain atmospheric conditions, the system's e-folding time is less than the simulation time step (300 s). As shown in Figure S14, results indicates that the e-folding time does not show a significant correlation with $J_{1.4}$. Under the majority of atmospheric conditions (77.3%), the nucleating system's e-folding time is less than 300 s. Instances where the e-folding time exceeds 300 s are primarily observed in winter clean conditions characterized by low temperature (T < ~270 K), low condensation sink (CS < ~0.003 s$^{-1}$), and low precursor concentrations (SA < ~$10^6$ cm$^{-3}$). These findings align with the observations of Olenius and Roldin (2022). It's important to emphasize that this e-folding time represents the duration required for the

system to transition from having only precursor molecules to reaching near-equilibrium concentrations of various clusters. In reality, cluster concentrations generally do not start from zero. Therefore, the calculated e-folding time serves as an upper limit estimate. Given the predominance of atmospheric conditions where the e-folding time falls within or below the simulation time step of 300 s, consequently, the steady-state treatment is generally deemed reasonable for our WRF-Chem/R2D-VBS simulations.

[Figure]

**Figure S14.** The variation of e-folding time with $J_{1.4}$ correlated with temperature (A), CS (B), SA concentration (C), and DMA concentration (D). The data points were calculated using a more sparse sequence of input parameters (T: 250, 260, 270, 280, 290, 300, 310, 320 (K); CS: $5.00 \times 10^{-4}$, $5.00 \times 10^{-3}$, $5.00 \times 10^{-2}$, $5.00 \times 10^{-1}$ (s$^{-1}$); SA: $1.00 \times 10^5$, $1.00 \times 10^6$, $1.00 \times 10^7$, $1.00 \times 10^8$ (cm$^{-3}$); DMA: $5.00 \times 10^6$, $5.00 \times 10^7$, $5.00 \times 10^8$ (cm$^{-3}$)) compared to those shown in Table S1.

We further investigated another common treatment that may introduce bias: neglecting cluster formation in consuming precursor during nucleation. Our examination focused on assessing the proportion of precursor consumption by cluster formation relative to precursor concentrations. As shown in Figure S15 and S16, we found that this proportion increases with $J_{1.4}$ for both SA and DMA. Under the majority of atmospheric conditions (82.0% for DMA and 57% for SA), proportions are below 10%. Proportions exceed 10% are predominantly observed in scenarios also characterized by low temperature (T < ~270 K) and low condensation sink (CS < ~0.003 s$^{-1}$), but with high deference in concentrations between DMA and SA. Specifically, elevated SA concentrations, which lead to significant DMA consumption through cluster formation, and vice versa, contribute to scenarios where precursor consumption by cluster formation exceeds 10%. It's noteworthy that our calculation of precursor consumption by cluster formation starts from zero cluster concentration. Also, in the real atmosphere, cluster concentrations are generally nonzero, leading to another upper limit estimate. Therefore, based on our analysis, it can be inferred that cluster

formation may not introduce significant bias into NPF simulations under typical atmospheric conditions. We have added this additional analysis and discussion of the potential impacts of these common treatments in NPF simulations to line 717-731 in the revised manuscript and the supporting information.

[Figure]

Figure S15. The variation of proportion of DMA consumption by cluster formation relative to precursor concentrations with $J_{1.4}$, correlated with temperature (A), CS (B), SA concentration (C), and DMA concentration (D). The input variables are consistent with Figure S14.

[Figure]

Figure S16. The variation of proportion of SA consumption by cluster formation relative to precursor concentrations with $J_{1.4}$, correlated with temperature (A), CS (B), SA concentration (C), and DMA concentration (D). The input variables are consistent with Figure S14.

Finally, I also encourage to refer to the ACDC code repository (https://github.com/tolenius/ACDC) in order to provide a reference for the model tools for reproducibility of simulation results.
**Response:** We have added the ACDC code repository in the revised manuscript in line 161.

**REFERENCES**

Li, Y., Shen, J., Zhao, B., Cai, R., Wang, S., Gao, Y., Shrivastava, M., Gao, D., Zheng, J., Kulmala, M., and Jiang, J.: A dynamic parameterization of sulfuric acid–dimethylamine nucleation and its application in three-dimensional modeling, Atmospheric Chemistry and Physics, 23, 8789-8804, 10.5194/acp-23-8789-2023, 2023.

Olenius, T. and Roldin, P.: Role of gas-molecular cluster-aerosol dynamics in atmospheric new-particle formation, Sci Rep, 12, 10135, 10.1038/s41598-022-14525-y, 2022.